# On the impact of activation and normalization in obtaining isometric embeddings at initialization

**Amir Joudaki**
ETH Zurich
amir.joudaki@inf.ethz.ch

**Hadi Daneshmand**
MIT/LIDS-FODSI
dhadi@mit.edu

**Francis Bach**
INRIA-ENS-PSL Paris
francis.bach@inria.fr

## Abstract

In this paper, we explore the structure of the penultimate Gram matrix in deep neural networks, which contains the pairwise inner products of outputs corresponding to a batch of inputs. In several architectures it has been observed that this Gram matrix becomes degenerate with depth at initialization, which dramatically slows training. Normalization layers, such as batch or layer normalization, play a pivotal role in preventing the rank collapse issue. Despite promising advances, the existing theoretical results do not extend to layer normalization, which is widely used in transformers, and can not quantitatively characterize the role of non-linear activations. To bridge this gap, we prove that layer normalization, in conjunction with activation layers, biases the Gram matrix of a multilayer perceptron towards the identity matrix at an exponential rate with depth at initialization. We quantify this rate using the Hermite expansion of the activation function.

## 1 Introduction

Optimization of deep neural networks is a challenging non-convex problem. Various components and optimization techniques have been developed over the last decades to make optimization feasible. Components such as activation functions [Hendrycks and Gimpel, 2016], normalization layers [Ioffe and Szegedy, 2015], and residual connections [He et al., 2016] have significantly influenced network training and have thus become the building blocks of neural networks. The practical success of these components has inspired extensive theoretical studies on the intricate role of weight initialization [Saxe et al., 2013, Daneshmand et al., 2021], normalization [Yang et al., 2019, Kohler et al., 2019, Daneshmand et al., 2021, 2023, Joudaki et al., 2023] and activation layers [Pennington et al., 2018, Joudaki et al., 2023], on neural network training. For example, the training of large language models hinges on carefully utilizing residual connections, normalization layers, and tailored activations [Vaswani et al., 2017, Radford et al., 2018]. Noci et al. [2022a] highlight that the absence or improper utilization of these components can substantially slow training.

To delve deeper into the influence of normalization and activation layers on training, one line of research has studied neural networks at initialization [Pennington et al., 2018, de G. Matthews et al., 2018, Jacot et al., 2018, Yang et al., 2019, Li et al., 2022]. Several studies have focused on the Gram matrix, which captures the inner products of intermediate representations for a batch of inputs, revealing that Gram matrices become degenerate as the network depth increases [Saxe et al., 2013, Daneshmand et al., 2021, Joudaki et al., 2023]. These issue of degeneracy or rank deficiency has been observed in multilayer perceptrons (MLPs) [Saxe et al., 2013, Daneshmand et al., 2020], convolutional networks [Bjorck et al., 2018], and transformers [Dong et al., 2021], posing challenges to the training process [Noci et al., 2022b, Pennington et al., 2018, Xiao et al., 2018]. Research

Code is available at: https://github.com/ajoudaki/deepnet-isometry

37th Conference on Neural Information Processing Systems (NeurIPS 2023).

indicates that normalization layers can act effectively to circumvent such Gram degeneracy, thereby improving training [Yang et al., 2019, Daneshmand et al., 2020, 2021, Bjorck et al., 2018].

Analyses based of neural networks in the mean-field, i.e., the infinite width regime, have revealed profound insights about initialization by characterizing the local solutions to the Gram dynamics Yang et al. [2019], Pennington et al. [2018]. However, these results do not guarantee global convergence towards the mean-field solutions or depend on technical assumptions that are challenging to verify numerically [Daneshmand et al., 2021, 2020, Joudaki et al., 2023]. Furthermore, all these theories primarily pertain to the network at initialization, where parameters are typically random and do not necessarily hold during or after the training. In this work, our objective is to bridge these existing gaps.

**Contributions.** Building upon existing literature that elucidates the spectral properties of the Gram matrix, we introduce the concept of isometry, which quantifies the similarity of the Gram matrix to the identity. Our initial theoretical finding demonstrates that isometry does not decrease under conditions of (batch and layer) normalization. This finding illuminates the bias of normalization layers towards isometry at various stages, namely initialization, during, and post-training.

We subsequently extend our analysis to explore the impact of non-linear activations on the isometry of intermediate representations in MLPs. Within the mean-field regime, we establish that non-linear activations incline the intermediate representations towards isometry at an exponential rate in depth. Our principal contribution is quantifying this rate by utilizing the Hermit polynomial expansion of activations. Intriguingly, our empirical experiments unveil a correlation between this rate and the convergence of stochastic gradient descent in MLPs equipped with layer normalization and standard activations used in practice.

## 2   Related works

A line of research investigates the interplay between signal propagation of the network and training. The existing literature postulates that in order to ensure fast training [Schoenholz et al., 2017, Poole et al., 2016], the network output must be sensitive to input changes, quantified by the spectrum of input-input Jacobean. This hypothesis is employed by Xiao et al. [2018] to train a 10,000-layer CNN using proper weight initialization without stabilizing components such as skip connection or normalization layers. He et al. [2023] demonstrate the critical role of the Jacobean spectra in large language models. In this paper, we analyze the spectrum of Gram matrices that connect to the spectral properties of input-output Jacobean.

Mean-field theory has been extensively used to characterize Gram matrix dynamics in the limit of infinite width. In this setting, the Gram matrix is a fixed point of a recurrence equation that depends on the network architecture [Schoenholz et al., 2017, Yang et al., 2019, Pennington et al., 2018]. This fixed-point analysis can provide insights into the structure and spectral properties of Gram matrices in deep neural networks, thereby shedding light on the degeneracy of Gram matrices in networks [Schoenholz et al., 2017, Yang et al., 2019]. However, often fixed-points are not unique, and they can be degenerate or non-degenerate Yang et al. [2019]. In this paper, we establish a convergence rate to a non-degenerate fixed-point for a family of MLPs.

Batch normalization [Ioffe and Szegedy, 2015] and layer normalization [Ba et al., 2016] layers are widely used in deep neural networks (DNNs) to improve training. Batch normalization ensures that each feature within a layer across a mini-batch has zero mean and unit variance. In contrast, layer normalization centers and divides the output of each layer by its standard deviation. There have been numerous theoretical studies on the effects of batch normalization due to its popularity Yang et al. [2019], Daneshmand et al. [2021], Joudaki et al. [2023]. While layer normalization has been the subject of increasing interest due to its application in transformers Xiong et al. [2020], there are relatively fewer studies on its theoretical underpinnings. While we primarily focus on layer normalization, we define and characterize a property that is shared between batch and layer normalization.

A broad spectrum of activation functions such as ReLU [Fukushima, 1969], GeLU [Hendrycks and Gimpel, 2016], SeLU [Klambauer et al., 2017], and Hyperbolic Tangent, and Sigmoid, are used in DNNs. These functions have various computational and statistical consequences in deep learning. Despite this diversity, only the design of SeLU activation is theoretically motivated [Klambauer et al.,

2017], while a broader theoretical understanding of activations remains elusive. To address this issue, we develop a theoretical framework to characterize the influence of a broad range of activations on intermediate representations in DNNs.

## 3   Preliminaries

**Notation.**   Let $\langle x, y \rangle$ be the inner product of vectors $x$ and $y$, and $\|x\|^2 = \langle x, x \rangle$ the squared Euclidean norm of $x$. For a matrix $X$, we write $X_{i\cdot}$ and $X_{\cdot i}$ for the $i$-th row and column of $X$, respectively. We use $W \sim N(\mu, \sigma^2)^{m \times n}$ to indicate that $W$ is an $m \times n$ Gaussian matrix with i.i.d. elements from $N(\mu, \sigma^2)$. We denote by $\mathbf{0}_n$ the zero vector of size $n$. Given vector $x \in \mathbb{R}^n$, $\overline{x}$ denotes the arithmetic mean of $\frac{1}{n} \sum_{i=1}^n x_i$. Lastly, $I_n$ is the identity matrix of size $n$.

**Normalization layers.**   Let $\text{LN} : \mathbb{R}^d \to \mathbb{R}^d$ and $\text{BN} : \mathbb{R}^{d \times n} \to \mathbb{R}^{d \times n}$, denote batch normalization and layer normalization respectively. Table 1 summarizes the definition of normalization layers. In our notations, we separate centering from normalization in layer (batch) normalization. Similarly, we split batch normalization into centering and normalization steps in our definitions. This notation allows us to decouple the effect of normalization from the centering. However, we will not depart from the standard MLP architectures as we include centering in the network architecture defined below.

| Width | $d \in \mathbb{N}$ | Batch size | $n \in \mathbb{N}$ |
|---|---|---|---|
| Depth | $L \in \mathbb{N}$ | Input | $x \in \mathbb{R}^d$ |
| Input batch | $X \in \mathbb{R}^{d \times n}$ | Gaussian weights | $W^1, \ldots, W^L \sim N(0,1)^{d \times d}$ |
| Activation | $\sigma : \mathbb{R} \to \mathbb{R}$ | Centering | $x - \overline{x}$ |
| **Layer Norm** | $\text{LN}(x) = \frac{x}{\sqrt{\frac{1}{d} \sum_i^d x_i^2}}$ | **Batch Norm** | $\text{BN}(X)_{ij} = \frac{X_{ij}}{\sqrt{\frac{1}{n} \sum_k^d X_{ik}^2}}$ |

Table 1: Building blocks we consider in this work.

**MLP setup.**   The subject of our analysis is an MLP with constant width $d$ across the layers and $L$ layers, which takes input $x \in \mathbb{R}^d$ and maps it to output $x^L \in \mathbb{R}^d$, with hidden representations as

$$\begin{cases} x^{\ell+1} = \frac{1}{\sqrt{d}} \text{LN}(h_\ell - \overline{h}_\ell), & h_\ell = \sigma(W^\ell x^\ell), \quad \ell = 0, \ldots, L-1 \\ x^0 := \frac{1}{\sqrt{d}} \text{LN}(x - \overline{x}), & \text{input.} \end{cases} \quad (1)$$

While the original ordering of layer normalization and activation is different [Ba et al., 2016], Xiong et al. [2020] show that the above ordering is more effective for large language models.

**Gram matrices and isometry.**   Given $n$ data points $\{x_i\}_{i \le n} \in \mathbb{R}^d$, the Gram matrix $G^\ell$ of the feature vectors $x_1^\ell, \ldots, x_n^\ell \in \mathbb{R}^d$ at layer $\ell$ of the network is defined as

$$G^\ell := \left[ \langle x_i^\ell, x_j^\ell \rangle \right]_{i,j \le n}, \qquad\qquad \ell = 0, 1, \ldots, L. \quad (2)$$

We define the notion of isometry to measure how much $G^\ell$ is close to a scaling factor of the identity matrix.

**Definition 1.**   *Let $G$ be an $n \times n$ positive semi-definite matrix. We define the* isometry $\mathcal{I}(G)$ *of $G$ as the ratio of its normalized determinant to its normalized trace:*

$$\mathcal{I}(G) := \frac{\det(G)^{1/n}}{\frac{1}{n} \text{tr}(G)}. \quad (3)$$

$\mathcal{I}(G^\ell)$ is a scale-invariant quantity measuring the parallelepiped volume spanned by the feature vectors $x_1^\ell, \ldots, x_n^\ell$. For example, consider two points on a plane $x_1, x_2 \in \mathbb{R}^2$ with lengths $a = |x_1|, b = |x_2|$ and angle $\theta = \angle(x_1, x_2)$. The ratio is given by $ab \sin(\theta)/(a^2 + b^2)$, which is maximized when $a = b$ and $\theta = \pi/2$. This relationship between volume and isometry is visually clear $n = 2$ and $n = 3$ feature vectors in Figure 1.

Remarkably, $\mathcal{I}(M)$ has the following properties (see Lemma A.1 for formal statements and proofs):

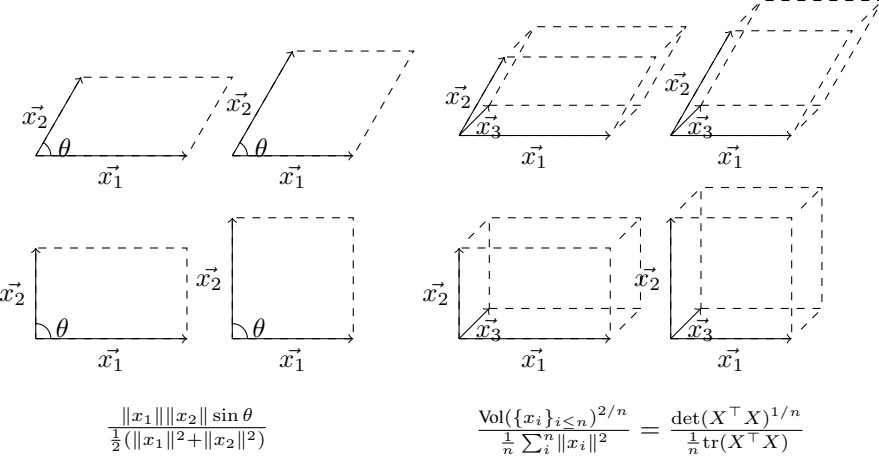

$$\frac{\|x_1\|\|x_2\|\sin\theta}{\frac{1}{2}(\|x_1\|^2+\|x_2\|^2)}$$

$$\frac{\mathrm{Vol}(\{x_i\}_{i\le n})^{2/n}}{\frac{1}{n}\sum_i^n\|x_i\|^2}=\frac{\det(X^\top X)^{1/n}}{\frac{1}{n}\mathrm{tr}(X^\top X)}$$

Figure 1: A geometric interpretation of isometry: higher volume, in the second row, corresponds to higher value for isometry.

(i) *Scaling-invariant:* For all constants $c > 0$, we have $\mathcal{I}(G) = \mathcal{I}(cG)$.

(ii) *Range:* $\mathcal{I} \in [0, 1]$ where the boundaries 0 and 1 are achieved for to degenerate and identity matrices respectively.

We also define the **isometry gap** as negative logarithm of isometry $-\log\mathcal{I}(M)$. Based on these properties of isometry, isometry gap lies between 0 and $\infty$, with 0 and $\infty$ indicating the perfect isometry (identity matrix) and degenerate matrices respectively. Isometry allows us to establish the inherent bias of normalization layers in the following section.

## 4 Isometry bias of normalization

This section is devoted to discussing the remarkable property of isometry in the context of normalization. We present a theorem that formalizes this property, followed by its geometric interpretation and implications.

**Theorem 1.** *Given $n$ samples $\{x_i\}_{i\le n} \subset \mathbb{R}^d \setminus \{\mathbf{0}_d\}$, their projection onto the unit sphere $\widetilde{x}_i := x_i/\|x_i\|$, and their respective Gram matrices $G$ and $\widetilde{G}$, the isometry obeys*

$$\mathcal{I}\left(\widetilde{G}\right) \ge \mathcal{I}(G)\left(1 + \frac{\frac{1}{n}\sum_i^n(a_i-\bar{a})^2}{\bar{a}^2}\right), \tag{4}$$

*where $a_i := \|x_i\|$, and $\bar{a} := \frac{1}{n}\sum_i^n a_i$.*

**Geometric Interpretation.** Isometry can be considered a measurement of the "volume" of the parallelepiped formed by sample vectors, made scale and dimension-independent. The normalization process effectively equalizes the edge lengths of this parallelepiped, enhancing the overall "volume" or isometry, provided there is a variance in the sample norms. Thus, projection onto the unit sphere $x_i \to x_i/\|x_i\|$ makes the edge lengths of the parallelepiped equal while leaving the angles between its edges intact. From this geometric perspective, Theorem 1 implies that among parallelepiped with similar angles between their edges and fixed total squared edge lengths, the one with equal edge lengths has the highest volume (and thereby isometry).

The proof of Theorem 1 is intuitive for the special case of vectors forming a cube, where max volume is realized when all edge lengths are equal. This fact that maximum volume is achieved when edge lengths are equal can be deduced from the arithmetic vs geometric mean inequality. Strikingly, the proof for the general case is nearly as simple as this special case. The high-level intuition behind the proof is that the determinant allows us to decouple the role of angles and edge lengths in volume formulation. This fact is evident for $n = 2$ in Figure 1. Since normalization does not modify the angles between edges, the remainder of the proof falls back onto the case where edges form a cube.

**Proof of Theorem 1.** Define $D := \text{diag}(a_1, \ldots, a_n)$. Observe that $G = D\widetilde{G}D$, implying $\det(G) = \det(\widetilde{G})\det(D)^2$. Because $\widetilde{x}_i$'s have norm 1, diagonals of Gram after normalization are constant $\widetilde{G}_{ii} = 1$, implying $\frac{1}{n}\text{tr}(\widetilde{G}) = 1$. We have

$$\frac{\mathcal{I}(\widetilde{G})}{\mathcal{I}(G)} = \frac{\frac{1}{n}\text{tr}(G)}{\frac{1}{n}\text{tr}(\widetilde{G})} \frac{\det(\widetilde{G})^{1/n}}{\det(G)^{1/n}} \tag{5}$$

$$= \frac{\frac{1}{n}\sum_i^n a_i^2}{1} \frac{\det(\widetilde{G})^{1/n}}{\det(\widetilde{G})^{1/n}(\prod_i^n a_i^2)^{1/n}} \tag{6}$$

$$= \frac{(\frac{1}{n}\sum_i a_i)^2}{(\prod_i^n a_i)^{2/n}} \frac{\frac{1}{n}\sum_i^n a_i^2}{(\frac{1}{n}\sum_i a_i)^2} \qquad\qquad \det(D) = \prod_i^n a_i^2 \tag{7}$$

$$\geq 1 \cdot \frac{\frac{1}{n}\sum_i^n a_i^2}{(\frac{1}{n}\sum_i^n a_i)^2} \qquad\qquad \frac{1}{n}\sum_i^n a_i \geq (\prod_i^n a_i)^{\frac{1}{n}} \tag{8}$$

$$= 1 + \frac{\frac{1}{n}\sum_i^n(a_i - \bar{a})^2}{\bar{a}^2}, \qquad\qquad \bar{a} := \frac{1}{n}\sum_i^n a_i. \tag{9}$$

$\square$

Theorem 1 further shows a subtle property of normalization: as long as there is some variation in the sample norms, i.e., $\|x_i\|$'s are not all equal, the post-normalization Gram has strictly higher isometry than the pre-normalization Gram matrix. It further quantifies the improvement in isometry as a function of variation of norms. Intuitively, terms $\bar{a}$ and $\frac{1}{n}\sum_i^n(a_i - \bar{a})^2$ can be interpreted as the average and variance of sample norms $a_1, \ldots, a_n$. Thus, a higher variation in the norms $a_i$'s leads to a larger increase in isometry after normalization.

### 4.1 Implications for layer (and batch) normalization

Theorem 1 reveals insights into the biases introduced by layer and batch normalization in neural networks, particularly highlighting the improvement in isometry not just limited to initialization but also persistent through the training process.

**Corollary 2.** *Consider $n$ vectors before and after layer-normalization $\{x_i\}_{i\leq n} \subset \mathbb{R}^d \setminus \{\mathbf{0}_d\}$ and $\{\widetilde{x}_i\}_{i\leq n}, \widetilde{x}_i := \text{LN}(x_i)$. Define their respective Gram matrices $G := [\langle x_i, x_j\rangle]_{i,j\leq n}$, and $\widetilde{G} := [\langle \widetilde{x}_i, \widetilde{x}_j\rangle]_{i,j\leq n}$. We have:*

$$\mathcal{I}\left(\widetilde{G}\right) \geq \mathcal{I}(G)\left(1 + \frac{\frac{1}{n}\sum_i^n(a_i - \bar{a})^2}{\bar{a}^2}\right), \qquad where\ a_i := \|x_i\|, \bar{a} := \frac{1}{n}\sum_i^n a_i.$$

What makes the above result distinct from related studies [Daneshmand et al., 2021, 2020, Yang et al., 2019] is that the increase in isometry is not limited to random initialization. Thus, layer normalization increases the isometry even during and after training. This calls for future research on the role of this inherent bias in enhanced optimization and generalization performance with batch normalization [Ioffe and Szegedy, 2015, Yang et al., 2019, Lyu et al., 2022, Kohler et al., 2019].

Despite the seemingly vast differences between layer normalization and batch normalization [Lubana et al., 2021], the following corollary shows a link between these two different normalization techniques.

**Corollary 3.** *Given $n$ samples in a mini-batch before $X \in \mathbb{R}^{d \times n}$, and after normalization $\widetilde{X} = \text{BN}(X)$ and define covariance matrices $C := XX^\top$ and $\widetilde{C} := XX^\top$. We have:*

$$\mathcal{I}\left(\widetilde{C}\right) \geq \mathcal{I}(C)\left(1 + \frac{\frac{1}{d}\sum_i^d(a_i - \bar{a})^2}{\bar{a}^2}\right), \qquad where\ a_i := \|X_{i\cdot}\|, \bar{a} := \frac{1}{n}\sum_{i=1}^d a_i.$$

Gram matrices of networks with batch normalization have been the subject of many previous studies at network initialization: it has been postulated that BN prevents rank collapse issue [Daneshmand

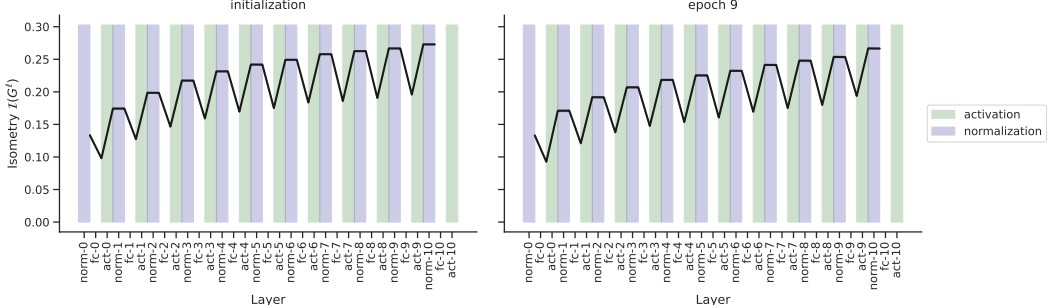

Figure 2: **Validation of Corollary 2** Isometry (y-axis) vs different layer of an MLP: Normalization layers (shaded blue) across all layers and configurations maintain or increase isometry both before (left) and after (right) training, validating Corollary 2. *Hyper parameters:* activation: $\tanh$, depth: 10, width: 1000, batch-size: 512, training SGD on training set of CIFAR10. with $lr = 0.01$.. Layer names are encoded as type-index, where type can be `fc`: fully connected, `norm`: LayerNorm, and `act`: activation.

et al., 2020] and that it orthogonalizes the representations [Daneshmand et al., 2021], and that it imposes isometry [Yang et al., 2019]. It is straightforward to verify that orthogonal matrices have the maximum isometry. Thus, the increase in isometry links to the orthogonalization of hidden representation characterized by Daneshmand et al. [2021]. While all previous results heavily rely on Gaussian random weights to establish this inherent bias, Corollary 3 is not limited to random weights.

### 4.2 Empirical validation of Corollary 2 in an MLP setup

We can validate Corollary 2 by tracking the isometry of various layers of an MLP with layer normalization. Figure 2 shows the isometry of intermediate representations in an MLP with layer normalization and hyperbolic tangent on CIFAR10 dataset. Shades in the figure mark layers illustrate that the isometry of the Gram matrix is non-decreasing after each layer normalization layer. We can see in Figure 2 that both before (left) and after training (right), the normalization layers maintain or improve isometry. To highlight the fact that the claims of Corollary 2 holds at all times and not only for initialization, Figure 2 tracks isometry of various layers both at initialization (left) and after training (right). This can be verified by the fact that isometry in the normalization layers (shaded blue) is either stable or increased, which validates Corollary 2.

## 5 Isometry bias of non-linear activation functions

So far, our focus was individual normalization layers. In this section, we extend our analysis to all layers when weights are **Gaussian**. Inspired by the isometry bias of normalization, we analyze how other components of neural networks influence the isometry of the Gram matrices, denoted by $\mathcal{I}(G^\ell)$ with a specific focus on non-linear activations.

### 5.1 Hermite expansion of activation functions

Analyzing Gram matrix dynamics for non-linear activation is challenging since even small modifications in the scale or shape of activations can lead to significant changes in the representations. A powerful tool to analyze activations is to express activations in the Hermite polynomial basis. Inspired by previous successful applications of Hermite polynomials in neural network analysis [Daniely et al., 2016, Yang, 2019], we explore their impact on the isometry of activation functions.

**Definition 2.** *Hermite polynomial of degree $k$, denoted by $He_k(x)$, is defined as*

$$He_k(x) := (k!)^{-\frac{1}{2}}(-1)^k e^{\frac{x^2}{2}} \frac{d^k}{dx^k} e^{-\frac{x^2}{2}}.$$

All square-integrable function $\sigma$ with respect to the Gaussian kernel, which obeys $\int_{-\infty}^{\infty} \sigma(x)^2 e^{-x^2/2} dx < \infty$, can be expressed as a linear combination of Hermite polynomials

as $\sigma(x) = \sum_k c_k He_k(x)$ with (see section A for more details):

$$c_k := \mathbb{E}_{x \sim \mathcal{N}(0,1)}[\sigma(x) He_k(x)].$$

The subsequent section will discuss how to leverage the Hermite expression of the activation to analyze the dynamics of Gram matrix isometry.

## 5.2 Non-linear activations bias Gram dynamics towards isometry

In this section, we analyze how $\mathcal{I}(G^\ell)$ changes with $\ell$. We use the mean-field dynamic of Gram matrices subject of previous studies [Yang et al., 2019, Schoenholz et al., 2017, Poole et al., 2016]. The mean-field dynamics of Gram matrices is given by

$$G_*^{\ell+1} = \mathbb{E}_{h \sim N(0, G_*^\ell)} \left[ \phi(\sigma(h))\phi(\sigma(h))^\top \right], \qquad \text{where } [\phi(a)]_i := (a_i - \mathbb{E}a_i)/\sqrt{\operatorname{Var} a_i}. \quad (10)$$

This equation gives the expected Gram matrix for layer $\ell + 1$, based on the Gram matrix $G_*^\ell$ from the previous layer, and $\phi$ is mean-field regime counterpart for layer normalization operator (see section A for more details). The sequence $G_*^\ell$ approximates the dynamics of $G^\ell$, and this correspondence becomes exact for infinitely wide MLPs. In the rest of this section, we analyze the above dynamical system. Our theory relies on the notion of *isometry strength* of the activation function, defined next.

**Definition 3** (Isometry strength). *Given activation $\sigma$ with Hermite expansion $\{c_k\}_{k \geq 0}$, define its isometry strength $\beta_\sigma$ as:*

$$\beta_\sigma := 2 - \frac{c_1^2}{\sum_{k=1}^\infty c_k^2} \quad (11)$$

We can readily check from the definition that isometry strength $\beta_\sigma$ has the following basic properties: (i) it ranges between 1 and 2, and (ii) it is 1 if and only if the activation is a linear function. Table 2 presents the isometry strength of certain activations in closed form. With this definition, we can finally analyze Gram matrix mean-field dynamics. Interestingly, the negative log of isometry $-\log \mathcal{I}(G_*^\ell)$ can serve as a Lyapunov function for the above dynamics. The following theorem proves non-linear activations also impose isometry similar to normalization layers.

**Theorem 4.** *Let $\sigma$ be an activation function with a Hermite expansion and a non-linearity strength $\beta_\sigma$, (see equation (11)). Given non-degenerate input Gram matrix $G_*^0$, then for sufficiently large layer $\ell \gtrsim \beta_\sigma^{-1}(-n \log \mathcal{I}(G_*^0) + \log(4n))$, we have*

$$-\log \mathcal{I}(G_*^\ell) \leq \exp(-\ell \log \beta_\sigma - n \log \mathcal{I}(G_*^0) + \log(4n)). \quad (12)$$

Note that the condition on input being non-degenerate is essential to reach isometry through depth. For example, if the input batch contains a duplicated sample, their corresponding representations across all layers will remain duplicated, implying that all $G_*^\ell$'s will be degenerate.

Theorem 4 reveals the importance of non-linear Hermite coefficients ($c_k, k \geq 2$) in activation function to ensure $\beta_\sigma > 1$ and obtain isometry in depth. This connection between $\beta_\sigma$ and isometry is the rationale for referring to $\beta_\sigma$ as the isometry strength. This constant can be computed in closed form for various activations, as shown in Table 2, and for all other activations, it can be computed numerically by sampling.

| | $He_1(x)$ | $He_2(x)$ | Sine | Exponential | Step | ReLU |
|---|---|---|---|---|---|---|
| $\sigma$ | $x$ | $x^2 - 1$ | $\sin(x)$ | $\exp(x-2)$ | $\mathbf{1}[x>0]$ | $\max(x,0)$ |
| $\beta_\sigma$ | 1 | 2 | $2 - \frac{2e}{e^2-1}$ | $2 - \frac{1}{e-1}$ | $2 - \frac{2}{\pi}$ | $\frac{3\pi-4}{2\pi-2}$ |

Table 2: Isometry strength $\beta_\sigma$ (see Definition 3) for various activation functions.

Figure 3 compares the established bound on the isometry gap with those observed in practice, i.e. $G^\ell$, for three activations. We observe $\beta_\sigma$ predicts the decay rate in isometry of Gram matrices $G^\ell$. While so far, we have only discussed the direct results of our theory for isometry of Gram matrices, in the next section, we will discuss other insights from the above analysis.

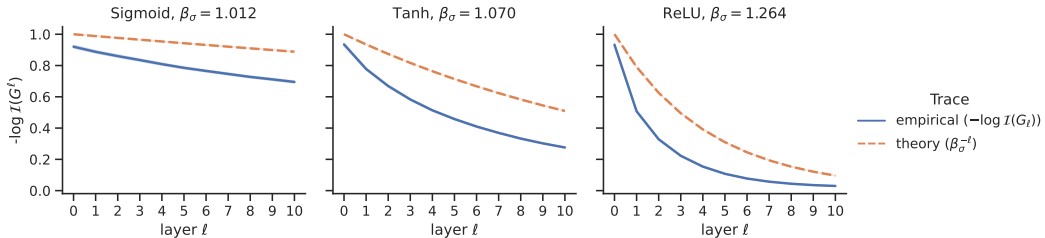

Figure 3: **Validation of Theorem 4**. Solid blue traces show the isometry of MLP with $n = 100$, $d = 5000$, and various activations $\sigma$. Solid lines show the average of #10 independent runs. The dashed traces are theoretical upper bounds given in Theorem 4, with constant $C = 1$.

## 6 Implications of our theory

In this section, we elucidate the implications of our theory, beginning with insights into layer normalization through the Hermit expansion of activation functions, followed by an examination of its impact on training.

Layer normalization primarily involves two steps: (i) centering and (ii) normalizing the norms. Through the Hermit expansion of activation functions, we unravel the underlying intricacies of these components and propose alternatives based on insights from the Hermit expansion.

**Experimental Setup.** Our experiments utilize MLPs with layer normalization and various activation functions for the task of image classification on the CIFAR10 dataset [Krizhevsky et al.]. For training, we use stochastic gradient descent (SGD) with a fixed step size of $0.01$. Unless stated otherwise, the MLP has constant width of $d = 1000$ is maintained across hidden layers, and a batch size of $n = 10$. Throughout this section, $a^\ell$ and $x^\ell$ respectively denote the post-activation and post-normalization vector for layer $\ell$.

### 6.1 Centering and Hermit expansion

One of the key insights of our mean-field theory is that the centering step in layer normalization is crucial in obtaining isometry. In the mean-field regime, pre-activations follow standard Gaussian distributions, and thus the average post-activation $\overline{a^\ell} = \frac{1}{d} \sum_i^d a_i^\ell$ will converge to their expectation $\overline{a^\ell} = \mathbb{E}_{z \sim \mathcal{N}(0,1)}[\sigma(X)] = c_0$. This insight suggests an alternative way of obtaining isometry by explicitly removing the offset term from activation, i.e., replacing activation $\sigma(x)$ by $\sigma(x) - c_0$. Strikingly, our experiments presented in Figure 4 indicate that such replacement can also impose isometry. This result provides novel insights into the role of centering in layer normalization.

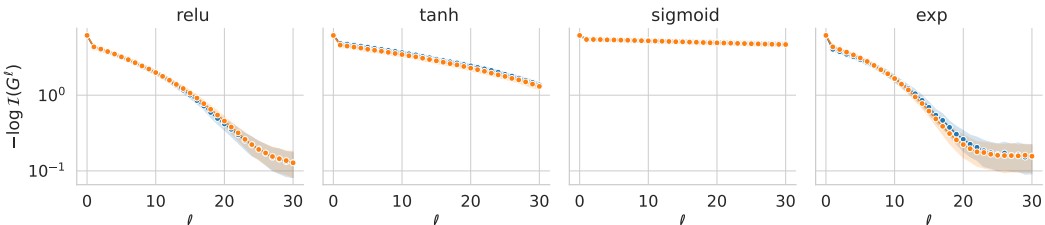

Figure 4: Layer vs. mean-field centering in obtaining isometry. Layer centering (orange): $x^{\ell+1} = \text{LN}(a^\ell - \overline{a^\ell})$. Mean-field centering (blue): $x^{\ell+1} = \text{LN}(a^\ell - c_0)$.

### 6.2 Normalization and Hermit expansion

Theorem 4 further reveals the importance of normalization of norms in addition to centering to achieve isometry. Figure 5 underlines the importance of the normalization for different activations, where we observe the isometry gap may increase without normalization. Similar to our mean-field analysis of centering, the factor $\frac{1}{d} \sum_{i=1}^d (a_i^\ell - \overline{a^\ell})^2$ in layer normalization converges to variance

$\text{var}_{z \sim N(0,1)}(\sigma(z)) = \sum_{k=1}^{\infty} c_k^2 =: \bar{\sigma}(1)$. Thus, as the width increases, the layer normalization operator $\text{LN}(a^\ell - \overline{a^\ell})$ will converge to $(a^\ell - c_0)/\sqrt{\bar{\sigma}(1)}$. Figure 6 demonstrates that the constant scaling $1/\sqrt{\bar{\sigma}(1)}$, achieves comparable isometry to layer normalization for hyperbolic tangent and sigmoid and ReLU, while it is not effective for $\exp$ function. This observation calls for future research on the link between normalization and activation in deep neural networks.

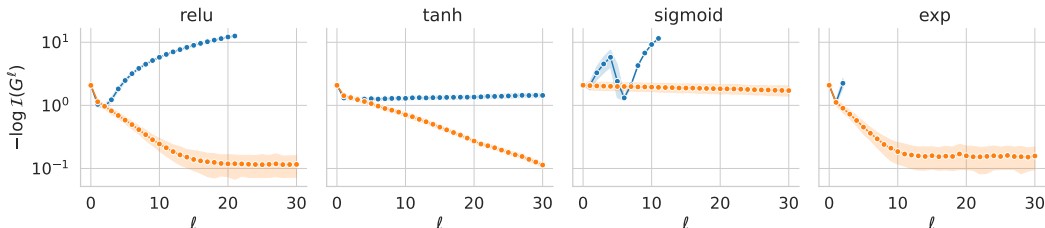

Figure 5: Comparing no normalization (blue) $x^{\ell+1} = (a^\ell - c_0)$ with layer normalization (orange) $x^{\ell+1} = \text{LN}(a^\ell - c_0)$ in obtaining isometry.

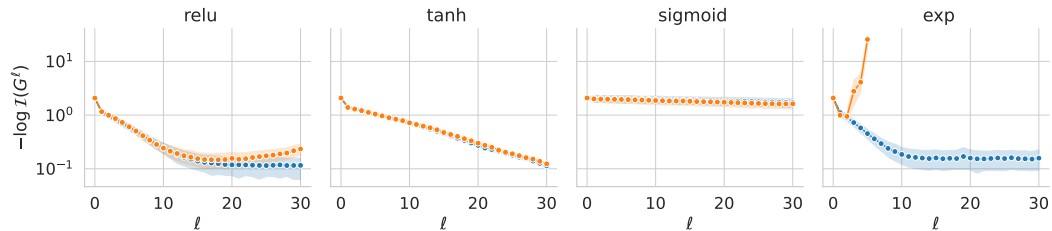

Figure 6: Comparing layer normalization (blue) $x^{\ell+1} = \text{LN}(a^\ell - c_0)$ vs. mean-field normalization (orange) $x^{\ell+1} = \frac{a^\ell - c_0}{\sqrt{\bar{\sigma}(1)}}$ in obtaining isometry.

### 6.3 Isometry strength correlates with SGD convergence rate in shallow MLPs.

Besides the direct consequences of our theory, we observe a striking correlation between the convergence of SGD and isometry strength in a specific range of neural network hyper-parameters. Figure 7 shows the convergence of SGD is faster for activations with a significantly larger isometry strength $\beta_\sigma$ (see Definition 3) for *shallow* MLPs, e.g., with 10 layers or less. We can speculate that this correlation reflects the input-output sensitivity of the networks with higher non-linearity. Surprisingly, this correlation does not extend to deeper networks. This discrepancy between shallow and deep networks regarding SGD convergence may be due to the issue of gradient explosion studied by Meterez et al. [2023]. This finding suggests multiple avenues for future research.

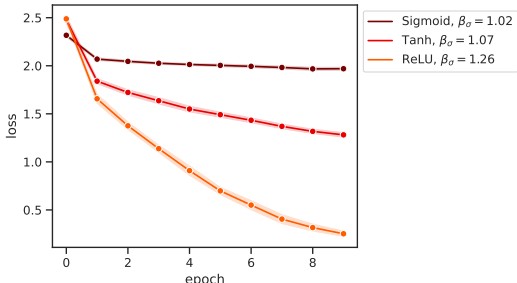

Figure 7: $\beta_\sigma$ **correlates with SGD training speed.** Training loss (y-axis) vs epoch (x-axis) on a training dataset of `CIFAR10`. Each curve shows the MLP with corresponding activation, with color codes denoting isometry strength $\beta_\sigma$. Hyper parameters: depth: 10, width: 1000, batch size: 512, Average of #5 independent runs.

# 7 Discussion

In this study, we explored the influence of layer normalization and nonlinear activation functions on the isometry of MLP representations. Our findings open up several avenues for future research.

**Self normalized activations.** It is worth investigating whether we can impose isometry without layer normalization. Our empirical observations suggest that certain activations, such as ReLU, require layer normalization to attain isometry. In contrast, other activations, which can be considered as "self-normalizing" (e.g., SeLU [Klambauer et al., 2017] and hyperbolic tangent), can achieve isometry with only offset and scale adjustments (see Figure 8). We experimentally show how we can replace centering and normalization by leveraging Hermit expansion of activation. Thus, we believe Hermit expansion provides a theoretical grounding to analyze the isometry of SeLU.

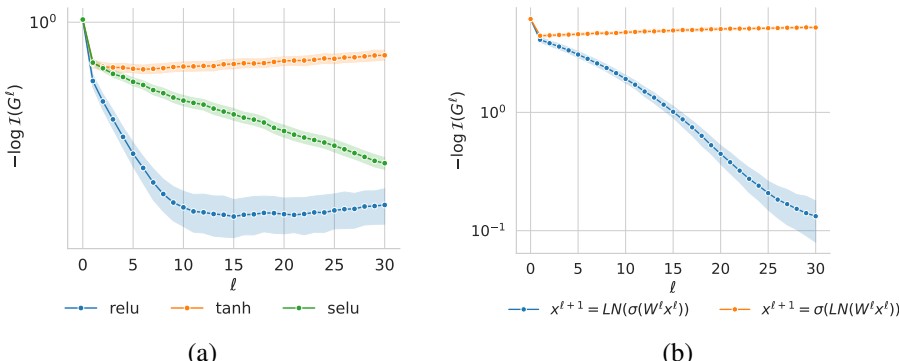

Figure 8: (a) Exploring the potential of achieving isometry with self-normalized activations. Batch size $n = 10$, width $d = 1000$. (b) Impact of the order of activation and normalization layers. Batch size $n = 10$, width $d = 1000$.

**Impact of the ordering of normalization and activation layers on isometry.** Theorem 4 highlights that the ordering of activation and normalization layers has a critical impact on the isometry. Figure 8 demonstrates that a different ordering can lead to a non-isotropic Gram matrix. Remarkably, the structure analyzed in this paper is used in transformers [Vaswani et al., 2017].

**Normalization's role in stabilizing mean-field accuracy through depth.** Numerous theoretical studies conjecture that mean-field predictions may not be reliable for considerably deep neural networks [Li et al., 2021, Joudaki et al., 2023]. Mean-field analysis incurs a $O(1/\sqrt{\text{width}})$ error per layer when the network width is finite. This error may accumulate with depth, making mean-field predictions increasingly inaccurate with an increase in depth. However, Figure 9 illustrates that layer normalization controls this error accumulation through depth. This might be attributable to the isometry bias induced by normalization, as proven in Theorem 1. Similarly, batch normalization also prevents error propagation with depth by imposing the same isometry [Joudaki et al., 2023]. This observation calls for future research on the essential role normalization plays in ensuring the accuracy of mean-field predictions.

# Acknowledgements

Amir Joudaki is funded through Swiss National Science Foundation Project Grant #200550 to Andre Kahles, and partially funded by ETH Core funding award to Gunnar Ratsch. Hadi Daneshmand acknowledges support from the NSF TRIPODS program (DMS-2022448).

# References

Dan Hendrycks and Kevin Gimpel. Gaussian error linear units (gelus). *arXiv preprint arXiv:1606.08415*, 2016.

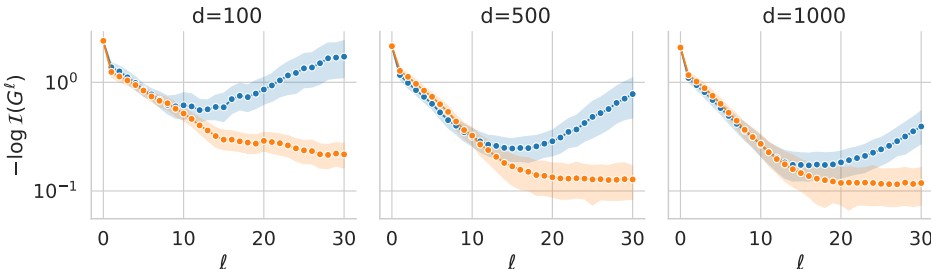

Figure 9: Role of normalization in stabilizing mean-field accuracy for ReLU, batch size $n = 10$. Mean-field normalization (blue): $x^{\ell+1} = \frac{a^\ell - c_0}{\sqrt{\bar{\sigma}(1)}}$. Layer normalization (orange): $x^{\ell+1} = \text{LN}(a^\ell - c_0)$

Sergey Ioffe and Christian Szegedy. Batch normalization: Accelerating deep network training by reducing internal covariate shift. In *International Conference on Machine Learning*, pages 448–456. pmlr, 2015.

Kaiming He, Xiangyu Zhang, Shaoqing Ren, and Jian Sun. Deep residual learning for image recognition. In *Proceedings of the IEEE conference on computer vision and pattern recognition*, pages 770–778, 2016.

Andrew M. Saxe, James L. McClelland, and Surya Ganguli. Exact solutions to the nonlinear dynamics of learning in deep linear neural networks. *arXiv preprint arXiv:1312.6120*, 2013.

Hadi Daneshmand, Amir Joudaki, and Francis Bach. Batch normalization orthogonalizes representations in deep random networks. *Advances in Neural Information Processing Systems*, 34: 4896–4906, 2021.

Greg Yang, Jeffrey Pennington, Vinay Rao, Jascha Sohl-Dickstein, and Samuel S Schoenholz. A mean field theory of batch normalization. *arXiv preprint arXiv:1902.08129*, 2019.

Jonas Kohler, Hadi Daneshmand, Aurelien Lucchi, Thomas Hofmann, Ming Zhou, and Klaus Neymeyr. Exponential convergence rates for batch normalization: The power of length-direction decoupling in non-convex optimization. In *The 22nd International Conference on Artificial Intelligence and Statistics*, pages 806–815. PMLR, 2019.

Hadi Daneshmand, Jason D Lee, and Chi Jin. Efficient displacement convex optimization with particle gradient descent. *International Conference on Machine Learning*, 2023.

Amir Joudaki, Hadi Daneshmand, and Francis Bach. On bridging the gap between mean field and finite width in deep random neural networks with batch normalization. *International Conference on Machine Learning*, 2023.

Jeffrey Pennington, Samuel Schoenholz, and Surya Ganguli. The emergence of spectral universality in deep networks. In *International Conference on Artificial Intelligence and Statistics*, pages 1924–1932, 2018.

Ashish Vaswani, Noam Shazeer, Niki Parmar, Jakob Uszkoreit, Llion Jones, Aidan N Gomez, Łukasz Kaiser, and Illia Polosukhin. Attention is all you need. *Advances in Neural Information Processing Systems*, 30, 2017.

Alec Radford, Karthik Narasimhan, Tim Salimans, Ilya Sutskever, et al. Improving language understanding by generative pre-training. 2018.

Lorenzo Noci, Sotiris Anagnostidis, Luca Biggio, Antonio Orvieto, Sidak Pal Singh, and Aurelien Lucchi. Signal propagation in transformers: Theoretical perspectives and the role of rank collapse. *Advances in Neural Information Processing Systems*, 35:27198–27211, 2022a.

Alexander G. de G. Matthews, Jiri Hron, Mark Rowland, Richard E. Turner, and Zoubin Ghahramani. Gaussian process behaviour in wide deep neural networks. In *International Conference on Learning Representations*, 2018.

Arthur Jacot, Franck Gabriel, and Clément Hongler. Neural tangent kernel: Convergence and generalization in neural networks. *Advances in Neural Information Processing Systems*, 31, 2018.

Mufan Bill Li, Mihai Nica, and Daniel M. Roy. The neural covariance sde: Shaped infinite depth-and-width networks at initialization. *Advances in Neural Information Processing Systems*, 2022.

Hadi Daneshmand, Jonas Kohler, Francis Bach, Thomas Hofmann, and Aurelien Lucchi. Batch normalization provably avoids ranks collapse for randomly initialised deep networks. *Advances in Neural Information Processing Systems*, 33:18387–18398, 2020.

Nils Bjorck, Carla P. Gomes, Bart Selman, and Kilian Q. Weinberger. Understanding batch normalization. *Advances in Neural Information Processing Systems*, 31, 2018.

Yihe Dong, Jean-Baptiste Cordonnier, and Andreas Loukas. Attention is not all you need: Pure attention loses rank doubly exponentially with depth. In *International Conference on Machine Learning*, pages 2793–2803, 2021.

Lorenzo Noci, Sotiris Anagnostidis, Luca Biggio, Antonio Orvieto, Sidak Pal Singh, and Aurelien Lucchi. Signal propagation in transformers: Theoretical perspectives and the role of rank collapse. *arXiv preprint arXiv:2206.03126*, 2022b.

Lechao Xiao, Yasaman Bahri, Jascha Sohl-Dickstein, Samuel Schoenholz, and Jeffrey Pennington. Dynamical isometry and a mean field theory of CNNs: How to train 10,000-layer vanilla convolutional neural networks. In *International Conference on Machine Learning*, pages 5393–5402, 2018.

Samuel S. Schoenholz, Justin Gilmer, Surya Ganguli, and Jascha Sohl-Dickstein. Deep information propagation. In *International Conference on Learning Representations*, 2017.

Ben Poole, Subhaneil Lahiri, Maithra Raghu, Jascha Sohl-Dickstein, and Surya Ganguli. Exponential expressivity in deep neural networks through transient chaos. *Advances in Neural Information Processing Systems*, 29, 2016.

Bobby He, James Martens, Guodong Zhang, Aleksandar Botev, Andrew Brock, Samuel L Smith, and Yee Whye Teh. Deep transformers without shortcuts: Modifying self-attention for faithful signal propagation. *arXiv preprint arXiv:2302.10322*, 2023.

Jimmy Lei Ba, Jamie Ryan Kiros, and Geoffrey E. Hinton. Layer normalization. *arXiv preprint arXiv:1607.06450*, 2016.

Ruibin Xiong, Yunchang Yang, Di He, Kai Zheng, Shuxin Zheng, Chen Xing, Huishuai Zhang, Yanyan Lan, Liwei Wang, and Tieyan Liu. On layer normalization in the transformer architecture. In *International Conference on Machine Learning*, pages 10524–10533, 2020.

Kunihiko Fukushima. Visual feature extraction by a multilayered network of analog threshold elements. *IEEE Transactions on Systems Science and Cybernetics*, 1969.

Günter Klambauer, Thomas Unterthiner, Andreas Mayr, and Sepp Hochreiter. Self-normalizing neural networks. *Advances in Neural Information Processing Systems*, 30, 2017.

Kaifeng Lyu, Zhiyuan Li, and Sanjeev Arora. Understanding the generalization benefit of normalization layers: Sharpness reduction. *Advances in Neural Information Processing Systems*, 35: 34689–34708, 2022.

Ekdeep S Lubana, Robert Dick, and Hidenori Tanaka. Beyond batchnorm: towards a unified understanding of normalization in deep learning. *Advances in Neural Information Processing Systems*, 34:4778–4791, 2021.

Amit Daniely, Roy Frostig, and Yoram Singer. Toward deeper understanding of neural networks: The power of initialization and a dual view on expressivity. *Advances in Neural Information Processing Systems*, 29, 2016.

Greg Yang. Scaling limits of wide neural networks with weight sharing: Gaussian process behavior, gradient independence, and neural tangent kernel derivation. *arXiv preprint arXiv:1902.04760*, 2019.

Alex Krizhevsky, Vinod Nair, and Geoffrey Hinton. Cifar-10 (canadian institute for advanced research). URL `http://www.cs.toronto.edu/~kriz/cifar.html`.

Alexandru Meterez, Amir Joudaki, Francesco Orabona, Alexander Immer, Gunnar Rätsch, and Hadi Daneshmand. Towards training without depth limits: Batch normalization without gradient explosion. *arXiv preprint arXiv:2310.02012*, 2023.

Mufan Li, Mihai Nica, and Dan Roy. The future is log-gaussian: Resnets and their infinite-depth-and-width limit at initialization. *Advances in Neural Information Processing Systems*, 34:7852–7864, 2021.

Semyon Aranovich Gershgorin. Uber die abgrenzung der eigenwerte einer matrix. *News of the Russian Academy of Sciences. Mathematical series*, (6):749–754, 1931.

F Gustav Mehler. Ueber die entwicklung einer function von beliebig vielen variablen nach laplaceschen functionen höherer ordnung. *Journal für die Reine und Angewandte Mathematik (in German)*, 1866.

Adam Paszke, Sam Gross, Francisco Massa, Adam Lerer, James Bradbury, Gregory Chanan, Trevor Killeen, Zeming Lin, Natalia Gimelshein, Luca Antiga, et al. Pytorch: An imperative style, high-performance deep learning library. *Advances in neural information processing systems*, 32, 2019.

Thomas Wolf, Lysandre Debut, Victor Sanh, Julien Chaumond, Clement Delangue, Anthony Moi, Pierric Cistac, Tim Rault, R'emi Louf, Morgan Funtowicz, Jamie Brew, and Guillaume Dulac-Arnold. Transformers: State-of-the-art natural language processing. *Proceedings of the 2020 Conference on Empirical Methods in Natural Language Processing: System Demonstrations*, pages 38–45, October 2020. URL `https://www.aclweb.org/anthology/2020.emnlp-demos.6`.

Xavier Glorot and Yoshua Bengio. Understanding the difficulty of training deep feedforward neural networks. In *Proceedings of the thirteenth international conference on artificial intelligence and statistics*, pages 249–256. JMLR Workshop and Conference Proceedings, 2010.

## Appendix outline

The appendix is partitioned into four main components, each serving its purpose as described:

1. Section A details the all the proofs, with most of it dedicated to proof of Theorem 4 alongside numerical confirmation of significant steps
   - Section A.1 elaborates on the basic properties of isometry.
   - Section A.2 provides an elaborate review of the mean-field Gram dynamics.
   - Section A.3 presents the Lyapunov function $\gamma$, and establishes that this function provides both upper and lower bounds for the isometry, thereby implying that geometric contraction of $\gamma(G^\ell)$ indicates a geometric contraction of isometry gap $-\log \mathcal{I}(G^\ell)$.
   - Section A.4 proves that $\gamma(G^\ell)$ exhibits an exponential contraction in depth with rate $\beta_\sigma$.

2. Section B outlines our rebuttal responses to the reviews that we chose to leave out of the main text.
   - Section B.1 gives additional details concerning the experiments reported in the main text and appendix.
   - Section B.2 explores the effect of gain on isometry, and links the rate to the associated isometry strength.
   - Section B.3 explores the effect of varying widths of hidden layers on the isometry.
   - Section B.4 explores the notion of isometry for representations in language models.

# A Proofs

## A.1 Basic properties of isometry

**Basic properties of isometry**  It is straightforward to check isometry obeys the following basic isometry-preserving properties:

**Lemma A.1.** *For PSD matrix $M$, the isometry defined in (3) obeys the following properties: 1) scale-invariance $\mathcal{I}(cM) = \mathcal{I}(M)$, 2) only takes value in the unit range $\mathcal{I}(M) \in [0,1]$ 3) it takes its maximum value if and only if $M$ is identity $\mathcal{I}(M) = 1 \iff M = I_n$, and 3) takes minimum value if and only if $M$ is degenerate $\mathcal{I}(M) = 0$.*

*Proof of Lemma A.1.* The scale-invariance is trivially true as scaling $M$ by any constant will scale $\det(M)^{1/n}$ and $\mathrm{tr}(M)$ by the same amount. The proof of other properties is a straightforward consequence of writing the isometry in terms of the eigenvalues $\mathcal{I}(M) = (\prod_i \lambda_i)^{1/n}/(\frac{1}{n}\sum_i \lambda_i)$, where $\lambda_i$'s are eigenvalues of $M$. By arithmetic vs geometric mean inequality over the eigenvalues we have $(\prod_i \lambda_i)^{1/n} \leq \frac{1}{n}\sum_i \lambda_i)$, which proves that $\mathcal{I}(M) \in [0,1]$. Furthermore, the inequality is tight iff the values are all equal $\lambda_1 = \cdots = \lambda_n$, which holds only for identity $M = I_n$. Finally, isometry is zero iff at least one eigenvalue is zero, which is the case for degenerate matrix $M$. $\qquad\square$

## A.2 Mean-field Gram Dynamics

Recall the mean-field Gram dynamics stated in equation (10):

$$G_*^{\ell+1} = \mathbb{E}_{h \sim N(0, G_*^\ell)}\left[\phi(\sigma(h))\phi(\sigma(h))^\top\right], \qquad \text{where } [\phi(a)]_i := (a_i - \mathbb{E}a_i)/\sqrt{\mathrm{Var}\, a_i}, \quad (13)$$

Assuming that inputs are encoded as columns of $X$, we can restate the MLP dynamics as follows

$$X^0 := \frac{1}{\sqrt{d}}\mathrm{LN}(X - \overline{X}), \qquad\qquad \text{inputs} \qquad\qquad (14)$$

$$H^\ell := W^\ell X^\ell, W^\ell \sim N(0,1)^{d \times d}, \qquad \text{preactivation} \qquad (15)$$

$$A^\ell := \sigma(H^\ell), \qquad\qquad \text{activations} \qquad\qquad (16)$$

$$X^{\ell+1} = \frac{1}{\sqrt{d}}\mathrm{LN}(A^\ell - \overline{A^\ell}) \qquad \text{centering \& normalization,} \qquad (17)$$

where centering and layer normalization are applied column-wise, as defined in the main text. Observe that Gram matrix of representations can be written as

$$G_d^{\ell+1} = X^{\ell+1^\top} X^{\ell+1} \qquad\qquad\qquad\qquad\qquad\qquad (18)$$

$$= \frac{1}{d}\sum_{k=1}^d \left(\frac{A_{k1}^\ell - \mu_1}{s_1}, \ldots, \frac{A_{kn}^\ell - \mu_n}{s_n}\right)^{\otimes 2}, \quad \mu_i := \frac{1}{d}\sum_{k=1}^d (A_{ki}^\ell), s_i := \sqrt{\frac{1}{d}\sum_{k=1}^d (A_{ki}^\ell - \mu_i)^2}, \quad (19)$$

where $\otimes$ denotes Hadamard product, and subscript $_d$ emphasises the dependence of Gram on width $d$. Note that conditioned on the previous layer, rows of $H^\ell$ and $A^\ell$ are i.i.d. , because of independence of rows of $W^\ell$. Thus, by law of large numbers, in the infinitely wide network regime, $\mu_i$ and $s_i$ will converge to the expected mean and variance respectively $\lim_{d\to\infty} \mu_i = \mathbb{E}A_{1i}^\ell$, and $\lim_{d\to\infty} s_i = \sqrt{\mathrm{Var}(A_{1i}^\ell)}$, for all $i = 1, \ldots, n$. By construction of $\phi$, in the infinitely wide regime, we can rewrite Gram dynamics as $\lim_{d\to\infty} G_d^\ell = \frac{1}{d} \sum_{k=1}^d \phi(A_{i\cdot}^\ell)^\top \phi(A_{i\cdot}^\ell)$. We can invoke the fact that rows of $A^\ell$ are i.i.d. to conclude that $G_d$ is the sample Gram matrix that converges to its expectation

$$\lim_{d\to\infty} G_d^\ell = \mathbb{E}\phi(A_{1\cdot}^\ell)\phi(A_{1\cdot}^\ell)^\top = \mathbb{E}_{h\sim N(0,G^\ell)}\phi(\sigma(h))\phi(\sigma(h))^\top =: G_*^\ell, \tag{20}$$

where $_*$ denotes the mean-field regime $d \to \infty$. This concludes the connection between the mean-field Gram dynamics and infinitely wide Gram dynamics.

## A.3 Introducing a potential

Here we will introduce a Lyapunov function that enables us to precisely quantify the isometry of activations in deep networks:

**Definition 4.** *Given a positive semidefinite matrix $G \in \mathbb{R}^{n\times n}$, we define $\gamma : \mathbb{R}^{n\times n} \to \mathbb{R}^{\geq 0}$ as:*

$$\gamma(G) := \max_{i\neq j} \frac{|\widetilde{G}_{ij}|}{1 - |\widetilde{G}_{ij}|}, \qquad\qquad \widetilde{G}_{ij} := G_{ij}/\sqrt{G_{ii}G_{jj}}. \tag{21}$$

Remarkably, $\gamma$ obey exhibits an geometric contraction under one MLP layer update, which is stated in the following theorem:

**Theorem A.2.** *Let $G$ be PSD matrix with unit diagonals $G_{ii} = 1, i = 1, \ldots, n$. It holds:*

$$\gamma\left(\mathbb{E}_{h\sim N(0,G)}\left[\phi(\sigma(h))\phi(\sigma(h))^\top\right]\right) \leq \gamma(G)/\beta_\sigma, \qquad \text{where } [\phi(a)]_i := (a_i - \mathbb{E}a_i)/\sqrt{\mathrm{Var}\,a_i}. \tag{22}$$

Thus, we may apply Theorem A.2 iteratively to prove that in the mean-field, the Lyapunov function $\gamma(G^\ell)$ decays at an exponential rate $\beta_\sigma$. A straightforward induction over layers leads to a decay rate in $\gamma$, which is presented in the next corollary.

**Corollary A.3.** *If activation $\sigma$ has Hermite expansion with non-linearity strength $\beta_\sigma$, the mean-field Gram matrices $G_*^\ell$, obey Lyapunov of these Gram matrices decays at an exponential rate $\beta_\sigma$:*

$$\gamma(G_*^\ell) \leq \gamma(G_*^0)\beta_\sigma^{-\ell}, \tag{23}$$

*where $G_*^0$ denotes the input Gram matrix.*

In Figure A.1, we experimentally validated the above equation for MLPs with a finite width and activations {tanh ,relu, sigmoid} where we observe that the mean-field analysis well predicates the decay in $\gamma$.

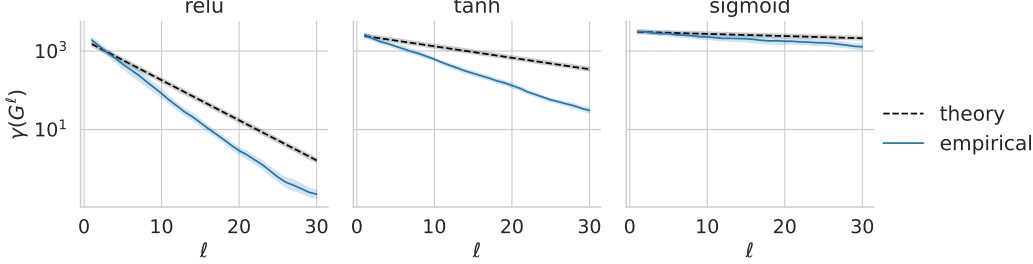

Figure A.1: $\gamma(G^\ell)$ vs depth $\ell$, for MLP with $n = 10$, $d = 10000$, various activations $\sigma$. Solid lines shows average of 10 independent runs. The dashed traces the theoretical upper bounds given in Corollary A.3.

Interestingly, we can connect the Lyapunov $\gamma$ to the isometry, by proving an upper and lower based on the determinant $G$ based on $\gamma(G)$, when $G$ is PSD and has unit diagonals:

**Lemma A.4.** *For PSD matrix $G$ with unit diagonals holds:*

$$\left(1 - (n-1)\max_{i\neq j}|G_{ij}|\right)^n \leq \det(G) \leq 1 - \max_{i\neq j} G_{ij}^2, \tag{24}$$

*where the lower bound holds if $(n-1)|G_{ij}| \leq 1$.*

Now, we are ready to prove the main theorem.

**Theorem A.5** (Restated Theorem 4). *Let $\sigma$ be an activation function with a Hermite expansion and a non-linearity strength $\beta_\sigma$, (see equation (11)). Given non-degenerate input Gram matrix $G_*^0$, then for sufficiently large layer $\ell \gtrsim \beta_\sigma^{-1}(-n\log\mathcal{I}(G_*^0) + \log(4n))$, we have*

$$-\log\mathcal{I}(G_*^\ell) \leq \exp(-\ell\log\beta_\sigma - n\log\mathcal{I}(G_*^0) + \log(4n)). \tag{25}$$

*Proof of Theorem 4.* First, note that we have the following transformation

$$t = \frac{x}{1-x} \implies x = \frac{t}{t+1} \implies \max_{i\neq j}|G_{ij}| = \frac{\gamma(G)}{1+\gamma(G)}, \tag{26}$$

we have

$$\det(G_*^\ell) \geq (1-(n-1)\max_{i\neq j}|G_{ij_*}^\ell|)^n \qquad \text{Lemma A.4} \tag{27}$$

$$\det(G_*^\ell) = \left(1-(n-1)(\gamma(G_*^\ell)/(1+\gamma(G_*^\ell)))\right)^n \qquad \text{Invoke (26)} \tag{28}$$

$$\geq \left(1-(n-1)\gamma(G_*^\ell)\right)^n \tag{29}$$

$$\implies -\frac{1}{n}\log\det(G_*^\ell) \leq -\log(1-(n-1)\gamma(G_*^\ell)) \tag{30}$$

$$\leq 2n\gamma(G_*^\ell) \qquad \text{for } \gamma(G_*^\ell) < 1/(2n-2) \tag{31}$$

$$\leq 2n\gamma(G_0)\beta_\sigma^{-\ell}. \tag{32}$$

By upper bound of Lemma A.4 we have $\max_{i\neq j}|G_{ij}| \leq \sqrt{1-\det(G)} \leq 1 - \det(G)/2$, implying $\gamma(G_0) \leq 2\det(G_0)^{-1}$, which allows us to

$$-\log\mathcal{I}(G_*^\ell) \leq 4n\det(G_0)^{-1}\beta_\sigma^{-\ell} \tag{33}$$

$$\leq \exp(-\ell\log\beta_\sigma - n\log\mathcal{I}(G_0) + \log(4n)) \tag{34}$$

which concludes the proof. $\qquad\square$

*Proof of Lemma A.4. Lower bound.* The lower bound is a result of Gershgorin circle theorem [Gershgorin, 1931], which implies that every eigenvalue must be within the disc $[1-(n-1)\max_{i\neq j}|G_{ij}|, 1+(n-1)\max_{i\neq j}|G_{ij}|]$. Thus, the determinant is lower-bounded by $(1-(n-1)\max_{i\neq j}|G_{ij}|)^n$.

*Upper bound.* Since $G$ is PSD, we can write $G = X^\top X$, which implies that columns of $X$ are unit norm $\|x_i\| = 1$, and $G$ encodes the angles between them $\cos\angle(x_i, x_j) = \langle x_i, x_j\rangle = G_{ij}$. Furthermore, we have $\det(G) = \text{vol}(X)^2$, where the volume refers to the parallelepiped spanned by the columns of $X$. With this formulation, we write volume recursively as volume spanned by columns 1 up to $n-1$, times the projection distance of $x_n$ from their span

$$\text{vol}(x_1, \ldots, x_n) = \text{dist}(x_n, \text{span}(x_1, \ldots, x_{n-1}))\text{vol}(x_1, \ldots, x_{n-1}),$$

where span refers to the space of all linear combinations of these vectors. Note that projection distance of $x_i$ onto $x_j$ can be written as $\sin\angle(x_i, x_j) = \sqrt{1-\langle x_i, x_j\rangle^2} = \sqrt{1-G_{ij}^2}$. Since the projection distance onto the linear span cannot be greater than projection distance onto a single vector, which is bounded by $\sqrt{1-\max_{i\neq j}G_{ij}^2}$. This concludes the upper bound that $\det(G) \leq 1-\max_{i\neq j}G_{ij}^2$. $\qquad\square$

## A.4 Dual activation and proof of Thm. A.2

According to Daniely et al. [2016], we leverage the notion dual activation associated with the activation function $\sigma$, which is defined in following.

**Definition 5.** *Given activation $\sigma : \mathbb{R} \to \mathbb{R}$ that is square integrable with respect to the Gaussian kernel, define its dual activation $\hat{\sigma} : \mathbb{R} \to \mathbb{R}$, and its mean reduced dual $\bar{\sigma} : \mathbb{R} \to \mathbb{R}$ as:*

$$\hat{\sigma}(\rho) := \mathbb{E}\sigma(x)\sigma(y) \qquad\qquad \bar{\sigma}(\rho) := \mathbb{E}\left[(\sigma(x) - \mathbb{E}\sigma(x))(\sigma(y) - \mathbb{E}\sigma(y))\right] \tag{35}$$

The following lemma connects the dual activation and its mean-reduced version with the Hermite expansion of $\sigma$:

**Lemma A.6.** *Given two standard Gaussian variables $X, Y \sim N(0, 1)$ with covariance $\mathbb{E}XY = \rho$, and activation $\sigma$ with normalized Hermite coefficients $c_{kk}$, we have*

$$\hat{\sigma}(\rho) = \sum_{k=0}^{\infty} c_k^2\rho^k, \qquad\qquad \bar{\sigma}(\rho) = \sum_{k=1}^{\infty} c_k^2\rho^k. \tag{36}$$

Finally, we have the tools to prove the first main theorem.

*Proof of Theorem A.2.* Let $a = \sigma(h)$ for $h \sim N(0, G)$. Note that by definition of the dual-activation we have $\mathbb{E}aa^\top = [\hat{\sigma}(G_{ij})]_{i,j \leq n}$. Since we assumed $G$ has unit diagonals, we have $h_i \sim N(0, 1)$, which implies that $\mathbb{E}a_i = \mathbb{E}_{h_i \sim N(0,1)}\sigma(h_i) = c_0$, implying that $\mathbb{E}(a_i - \mathbb{E}a_i)(a_j - \mathbb{E}a_j)^\top = \hat{\sigma}(G_{ij}) - c_0^2 = \bar{\sigma}(G_{ij})$. Furthermore, the variance can be driven as $\text{Var}(a_i) = \mathbb{E}a_i^2 - (\mathbb{E}a_i)^2 = \sum_{k=0}^{\infty} c_k^2 = \sum_{k=1}^{\infty} c_k^2 = \bar{\sigma}(1)$ for all $i = 1, \ldots, n$. Thus, we have $\mathbb{E}\phi(a_i)\phi(a_j) = \bar{\sigma}(G_{ij})/\bar{\sigma}(1)$. In the matrix form we have

$$\gamma\left(\mathbb{E}_{h \sim N(0,G)}\left[\phi(\sigma(h))\phi(\sigma(h))^\top\right]\right) = \gamma\left([\bar{\sigma}(G_{ij})/\bar{\sigma}(1)]_{i,j \leq n}\right) \tag{37}$$

The remainder proof relies on the following contractive property of Gram matrix potential:

**Lemma A.7.** *Consider activation $\sigma$, with normalized Hermite coefficients $\{c_k\}_{k \geq 0}$. For all $\rho \in (0, 1)$, the mean-reduced dual activation $\bar{\sigma}$ obeys*

$$\frac{|\bar{\sigma}(\rho)|/\bar{\sigma}(1)}{1 - |\bar{\sigma}(\rho)|/\bar{\sigma}(1)} \leq \beta_\sigma^{-1}\frac{|\rho|}{1 - |\rho|}, \tag{38}$$

*which the right hand-side is strictly larger if some nonlinear coefficient is nonzero $c_k \neq 0$ for some $k \geq 2$.*

Thus we can apply Lemma A.7 on each element $i \neq j$ to conclude that

$$\frac{|\bar{\sigma}(G_{ij})|/\bar{\sigma}(1)}{1 - |\bar{\sigma}(G_{ij})|/\bar{\sigma}(1)} \leq \frac{|G_{ij}|}{1 - |G_{ij}|}\beta_\sigma^{-1} \qquad \text{Lemma A.7 for all } i \neq j \tag{39}$$

$$\leq \beta_\sigma^{-1}\max_{i \neq j}\frac{|G_{ij}|}{1 - |G_{ij}|} \tag{40}$$

$$= \beta_\sigma^{-1}\gamma(G). \tag{41}$$

$$\tag{42}$$

since the inequality holds for any value of $i \neq j$, we can take the maximum over $i \neq jj$ to write:

$$\gamma\left(\bar{\sigma}(G)/\bar{\sigma}(1)\right) = \max_{i \neq j}\frac{|\bar{\sigma}(G_{ij})|/\bar{\sigma}(1)}{1 - |\bar{\sigma}(G_{ij})|/\bar{\sigma}(1)} \leq \beta_\sigma^{-1}\gamma(G), \tag{43}$$

which concludes the proof. $\qquad \square$

*Proof of Lemma A.7.* Note the ratio is invariant to scaling of $\sum_{k=1}^{\infty} c_k^2$. Hence, we assume $\sum_{k=1}^{\infty} c_k^2 = 1$ without loss of generality. With this simplification, we have $\bar{\sigma}(1) = 1$. For the positive range $\rho \in [0, 1]$ we have

$$\left(\frac{\bar{\sigma}(\rho)}{1 - \bar{\sigma}(\rho)}\right)\left(\frac{\rho}{1 - \rho}\right)^{-1} = \frac{\rho^{-1}\sum_{k=1}^{\infty} c_k^2 \rho^k}{(1 - \rho)^{-1}\sum_{k=1}^{\infty} c_k^2(1 - \rho^k)} \tag{44}$$

$$= \frac{\sum_{k=1}^{\infty} c_k^2 \rho^{k-1}}{\sum_{k=1}^{\infty} c_k^2\left(\frac{1 - \rho^k}{1 - \rho}\right)} \tag{45}$$

$$= \frac{\sum_{k=1}^{\infty} c_k^2 \rho^{k-1}}{\sum_{k=1}^{\infty} c_k^2\left(\sum_{i=0}^{k-1} \rho^i\right)} \tag{46}$$

$$\leq \frac{\sum_{k=1}^{\infty} c_k^2 \rho^{k-1}}{\left(\sum_{k=1}^{\infty} c_k^2 \rho^{k-1}\right) + \sum_{k=2}^{\infty} c_k^2} \tag{47}$$

$$\leq \max_{\rho \in [0,1]}\frac{\sum_{k=1}^{\infty} c_k^2 \rho^{k-1}}{\left(\sum_{k=1}^{\infty} c_k^2 \rho^{k-1}\right) + \sum_{k=2}^{\infty} c_k^2} \tag{48}$$

$$= \frac{\sum_{k=1}^{\infty} c_k^2}{\sum_{k=1}^{\infty} c_k^2 + \sum_{k=1}^{\infty} c_k^2 - c_1^2} \tag{49}$$

$$= \frac{1}{\beta_\sigma}. \tag{50}$$

Thus for $\rho \in [0, 1]$ we have

$$\frac{\bar{\sigma}(\rho)}{1 - \bar{\sigma}(\rho)} \leq \frac{\rho}{1 - \rho}\beta_\sigma^{-1}. \tag{51}$$

By Jensen inequality for convex function $x \mapsto |x|$ we have

$$|\bar{\sigma}(\rho)| = |\sum_{k=1}^{\infty} c_k^2 \rho^k| \leq \sum_{k=1}^{\infty} c_k^2|\rho|^k = \bar{\sigma}(|\rho|) \tag{52}$$

Because $x \mapsto x/(1-x)$ is monotonically increasing for $x \in [0, 1]$, we have

$$\frac{|\bar{\sigma}(\rho)|}{1 - |\bar{\sigma}(\rho)|} \leq \frac{\bar{\sigma}(|\rho|)}{1 - \bar{\sigma}(|\rho|)} \leq \frac{|\rho|}{1 - |\rho|} \beta_\sigma^{-1}. \tag{53}$$

Where we invoked the inequality that was proven for $\rho \in [0, 1]$, because $|\rho| \in [0, 1]$. $\qquad\square$

The following lemma, which is a consequence of Mehler's formula, is at the hart of proof of Lemma A.6:

**Lemma A.8** (Consequence of Mehler's kernel). *If $X, Y \sim N(0,1)$ with covariance $EXY = \rho$ we have*

$$E_{X,Y} He_n(X) He_k(Y) = \rho^n \delta_{nk}$$

*where $\delta_{nk}$ is the Dirac delta.*

*Proof of Lemma A.6.* Let $\sigma(x) = \sum_{k=0} c_k He_k(x)$, denote the Hermite expansion of $\sigma$. Thus, we have

$$\hat{\sigma}(\rho) := \mathbb{E}_{X,Y} \sigma(X) \sigma(Y) \tag{54}$$

$$= \sum_{n,k=0}^{\infty} c_n c_k \mathbb{E}_{X,Y} He_n(X) He_k(Y) \tag{55}$$

$$= \sum_{k=0}^{\infty} c_k^2 \rho^k, \tag{56}$$

where in the last line we applied result of Lemma A.8. For the mean-reduced dual kernel $\bar{\sigma}$, observe that $\mathbb{E}_{X \sim N(0,1)} \sigma(X) = c_0$. Thus, the reduction of mean will cancel the $k = 0$ term, which concludes the proof that $\bar{\sigma}(\rho) = \sum_{k=1} c_k^2 \rho^k$. $\qquad\square$

*Proof of Lemma A.8.* The property can be deduced from Mehler's formula Mehler [1866]. The formula states that

$$\frac{1}{\sqrt{1 - \rho^2}} \exp\left( -\frac{\rho^2(x^2 + y^2) - 2xy\rho}{2(1 - \rho^2)} \right) \tag{57}$$

$$= \sum_{m=0}^{\infty} He_m(x) He_m(y), \tag{58}$$

where the $m!$ factor difference is due to the definition of Hermite polynomials with an additional $1/\sqrt{m!}$ compared to the one used in Mehler's kernel. Observe that the left hand side is equal to $p(x, y)/p(x)p(y)$, where $p(x, y)$ is the joint PDF of $(X, Y)$, and $p(x), p(y)$ are PDF of $X$ and $Y$ respectively. Therefore, we can take the expectation using the expansion

$$\mathbb{E}_{X,Y}\left[ He_n(X) He_k(Y) \right] = \int He_n(x) He_k(y) p(x, y) dx dy \tag{59}$$

$$= \sum_{m=0}^{\infty} \rho^m \int He_n(x) He_k(y) He_m(x) He_m(y) dp(x) dp(y) \tag{60}$$

$$= \sum_{m=0}^{\infty} \rho^m \mathbb{E}_{X \sim N(0,1)}\left[ He_n(X) He_m(X) \right] \mathbb{E}_{Y \sim N(0,1)}\left[ He_k(Y) He_m(Y) \right] \tag{61}$$

$$= \rho^n \delta_{nk} \tag{62}$$

where in the last line we used the orthogonality property $E_{X \sim N(0,1)} H_k(x) H_n(X) = \delta_{nk}$. $\qquad\square$

# B  Additional experiments

## B.1  Details about empirical validations

**Hardware.** Experiments that did not require training a model were run on a `AMD Ryzen 9 3900X 12-Core` CPU, which takes about 5 minutes overall. All experiments that required training a model were trained a single `NVIDIA GeForce RTX 3090` GPU, which takes about 1 minutes per each training task.

**Figures.** The solid lines in all plots represent the average performance over multiple independent runs, and the shaded regions indicate the confidence intervals. Unless stated otherwise, each average is computed over #10 independent runs.

**Codes and reproducibility.** We implemented our experiments in Python using the PyTorch framework Paszke et al. [2019]. All the figures are reproducible with the code attached in the supplementary.

**Training procedure** For all training-related experiments, the isometry or isometry gap are computed per each batch by sampling a few of the batches randomly, and then averaged over. The epoch $i$ corresponds to the network at after $i$ steps of training on the training set of CIFAR10 (epoch 0 means network is at initialization).

**Pre-trained large language models** The pre-trained language models and their default configuration was downloaded from Huggingface Wolf et al. [2020] library.

## B.2 Quantifying the influence of gain on isometry through non-linearity strength

The concept of gain in neural networks is vital and closely connected with the weights initialization. A neural network with properly initialized weights can learn faster, have a lesser chance of getting stuck at sub-optimal solutions, and provide better generalization. The impact of gain can be visualized through the lens of weight initialization strategies such as Xavier normalization [Glorot and Bengio, 2010], which has shown significant effectiveness in optimizing neural networks. These initialization strategies apply a gain value to the weights, which is a scaling factor, to ensure a good signal flow through many layers during the forward and backward passes. The gain value essentially determines the variance of the weights in the initialization stage.

As an extension to our prior investigations, we delve into understanding the influence of gain on isometry, predominantly through our calculated metric, the non-linearity strength, denoted as $\beta_\sigma$, as a function of gain $\alpha$. For certain instances, such as ReLU, sine, and exponential activations, we are capable of deriving $\beta_\sigma(\alpha)$ in a closed form. Table B.1 presents a few of these cases.

| $\sigma$ | $\exp(\alpha x)$ | $\sin(\alpha x)$ | $\max(\alpha x, 0)$ |
|---|---|---|---|
| $\beta_\sigma$ | $\dfrac{2 - 2e^{\alpha^2} + \alpha^2}{1 - e^{\alpha^2}}$ | $\dfrac{2(-1 + e^{2\alpha^2} - e^{\alpha^2}\alpha^2)}{-1 + e^{2\alpha^2}}$ | $\dfrac{4 - 3\pi}{2 - 2\pi}$ |

Table B.1: Relationship between non-linearity strength and gain.

For a more extensive selection of activation functions, we have numerically computed the non-linearity strength as a function of gain $\alpha$, as visualized in Figure B.1. Leveraging Theorem 4, these computed values provide an estimation for the isometry strength for various activations. This correlation proves to be remarkably predictive, as shown in Figure B.2. Remarkably, in the case of ReLU activation, its unique characteristics lead to both our closed form $\beta_\sigma$ (refer Table B.1) and rate towards isometry (refer Figure B.2) remaining consistent across different values of $\alpha$.

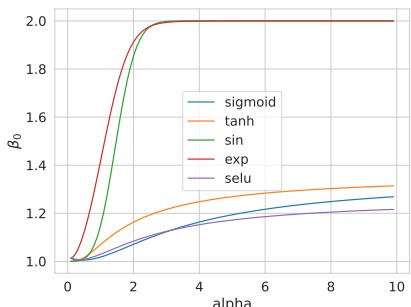

Figure B.1: Analysis of gain through the lens of isometry strength.

**Comparison to Xavier gain for initialization** Inspired by the results so far, we can compare mean-field centering and normalization to Xavier gain for activations. Figure B.3 demonstrates that all mean-field based gains improve the isometry when compared with Xavier initialization. However, this is markedly stronger for ReLU and leaky ReLU. We can explain this starker contrast by the fact that both activations have a significant offset term $c_0$, which is not corrected by the Xavier initialization.

## B.3 Varying width of hidden layers

While in our theoretical setup, we assume the network width is constant across the layers, this is only a choice to streamline our proof and notation. Since our primary result is derived from the mean-field regime, the

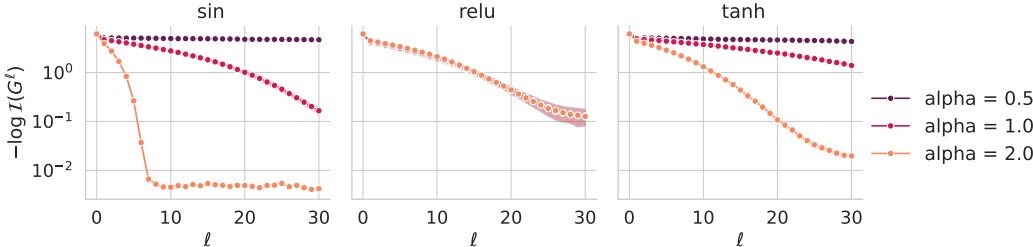

Figure B.2: Influence of gain on the attainment of isometry.

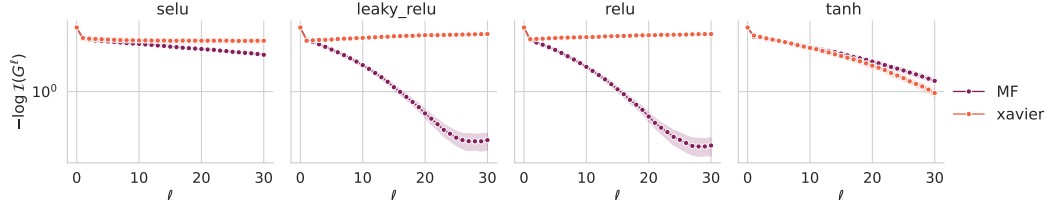

Figure B.3: Isometry vs depth for mean-field centering and normalization to Xavier initialization.

only criterion for it to hold is for the width to be sufficiently large to approximate the mean-field regime. Our experiments in Figure B.4 substantiate this claim that the specific sizes of hidden layers, as long as they are large, will not impact our main results on the isometry. We empirically validate this for four different configurations and show that the decay of the isometry gap remains largely consistent across these configurations.

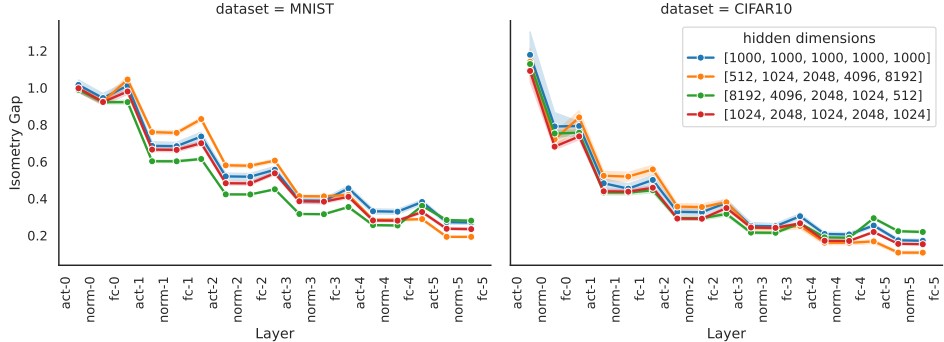

Figure B.4: **Theorem 4 holds for variable width MLP**. Isometry gap of various layers for MLP with ReLU activation and fixed width (blue), growing width (orange), narrowing width (green), and interchanging width (red) on MNIST (left) and CIFAR10 (right) training data. The weights of MLP are random. These plots show that our central theory holds regardless of the dataset or shape of MLP widths.

## B.4  Isometry in pre-trained large language models

Since our theory for normalization is not limited to initialization, we can expand our search for isometry to other architectures. Figure B.5 shows the important role of normalization in the pre-trained GPT2 network. However, we need to adjust the notion of isometry with the architecture of layer norm in a transformer in mind. In fact, the mean and standard deviation are computed over features separately for each token. Thus, to adapt the notion of isometry, we can view each token as a sample and define Gram over different tokens. Thus, isometry here quantifies the similarity between various tokens within one sample. As can be seen in the figure below, LayerNorm layers (shaded in red) in the last six layers of the pre-trained GPT2 increase the isometry between tokens, which is consistent with our theory of layer normalization. It is crucial that our theory holds deterministically, which extends to the pre-trained model.

One caveat in interpreting our results is that, in practice, LayerNorm layers have learnable parameters that make them deviate from our theory. It would be fruitful to study the effects of learned parameters to discern it from the role of centering and normalization for a future study.

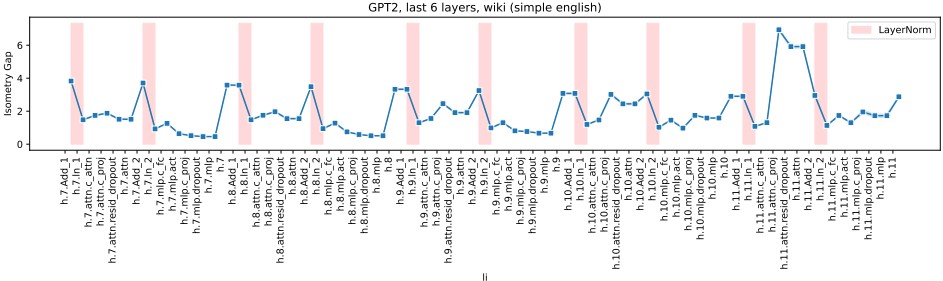

Figure B.5: **Validation of corollary 2 in GPT2**. The last six transformer layers of pre-trained GPT2, LayerNorm layers (shaded; red) decrease the isometry gap.

## B.5    Tracking isometry at initialization and optimization for more activations

Here there are more numerical experiments related to to tracking isometry before and after training.

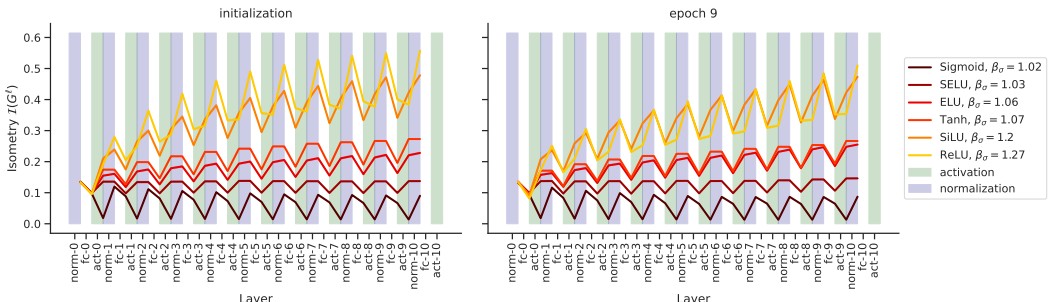

Figure B.6: Validation of Corollary 2 and Theorem 4 for multiple activations.

The following plot shows that isometry gap remains relatively stable.

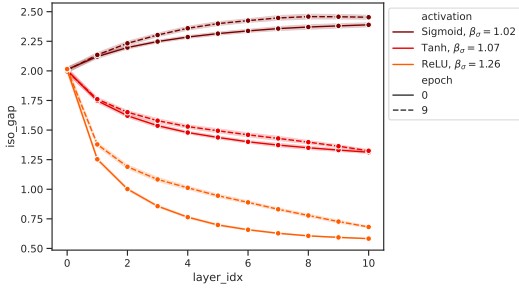

Figure B.7: Isometry gap change during training is relatively small.