# OpenReview forum: "On the impact of activation  and normalization in obtaining  isometric embeddings at initialization"
_NeurIPS.cc/2023/Conference — NeurIPS 2023 poster_

### Official Review · Reviewer_B3t5 · 2023-07-03

**Soundness:** 4 excellent
**Presentation:** 2 fair
**Contribution:** 4 excellent
**Rating:** 7
**Confidence:** 3

**Summary:**

A study of layernorm and Gramm matrix isometry is presented in both theory and emprical results. The results explain effectiveness of transformer (and similar) architechtures. Application of layernorm after activation seems to be a correct strategy when we are interested to isometry of output Gramm matrix.

**Strengths:**

Good theoretical contributions in the field of explaining the effect of layernorm in deep neural networks via the concept of Gramm matrix isometry. Authors also added a large number of illustrative empirical results that help to drive home the main points of the paper.

**Weaknesses:**

My main issue with the paper is not with the theory, but how it is presented in the main paper. The appendices appear to be fairly thorough mathematical exposition, but same rigorous exposition is absent in the main paper. Main paper should be readable without referencing to the appendices.

Here under, I itemize places where I feel more rigour in exposition would be needed:
From Theorem 1: "in isometry as a function of fluctuations in norms". What specifically is ment by "fluctuations in norms"?

Isn't this statement obvious "corollary 2 shows that the isometry of layer normalization is deterministic and does not rely on the random weights." ? In the setup of Corollary 2, no weight matrix is applied. So how this question even would be raised? Completely different issue is that if you would compare
I(G_l) and I(G_{l+1}), there weight matrix and activation function are applied. Next sentence about empirical proof also is out of place in this context.

In Section 3.1. what is ment by the sentence "which implies orthogonalization of these results."? Orthogonalization of what?

It would be good to expand the properties of Def 3 a bit more. Now we only have that it is "non-linearity strength for activations". Paper has half a page space for extra content. I would appreciate a few paragraphs of extra content in this context.

About Eq. (7), you mention that G_*^l is a sequence that approximates the G^l. From these paragraphs it is hard to see where the sequence is coming from.

**Questions:**

Gap between theory (Theorem 4) and observed in Fig. 4 is quite big. As a future work, is there a clear path for obtaining tighter bound?

**Limitations:**

-

---

> ### Author Rebuttal · Authors · 2023-08-10
>
>
> We would like to thank the reviewer for their positive review on our work
>
>
> > My main issue with the paper is not with the theory, but how it is presented in the main paper. ... some rigorous exposition is absent in the main paper. Main paper should be readable without referencing to the appendices.
>
> We thank the reviewer for these constructive feedback and will address it in the revised manuscript.
>
> > ... From Theorem 1: "in isometry as a function of fluctuations in norms". What specifically is ment by "fluctuations in norms"?
>
> To address this ambiguity we will replace "fluctuation of norms" by "variations"
>
> > - Isn't this statement obvious "corollary 2 shows that the isometry of layer normalization is deterministic and does not rely on the random weights." ? In the setup of Corollary 2, no weight matrix is applied. So how this question even would be raised? Completely different issue is that if you would compare $I(G_l)$ and $I(G_{l+1})$, there weight matrix and activation function are applied. Next sentence about empirical proof also is out of place in this context.
>
> This is a great and nuanced observation. We agree with the reviewer that the current phrasing does seem ambiguous. These two paragraphs (lines 124-130) stress that the non-decreasing isometry under `Layer Norm` is not a probablistic statement, but holds at all times.
>
> In order to track and analyze evolution of isometry through depth, we are fundumentally faced with the following chain $\dots \to x^l \to h^l \to x^{l+1} \to \dots$ where $h^{l} = \sigma(W^l \tilde{x}^l)$ and $x^{l+1} = \text{LayerNorm}(h^l).$ since $x^l$'s and $h^l$'s clearly depend on the preceding weights, one may attempt to anlyze their isometry by relying on the stochasticity of weights, and perhaps on their particular distribution (namely Gaussian weights.) Corrollary 2 and the following paragraphs assert that this isometry-preserving property does not rely on such stochastic nor mean-field approximations and holds at all times, implying that isometry of hidden representations $x_l$ cannot be smaller than isometry of hidden activations $h^l$.
>
> The empirical evidence in the attached PDF to general response solidifies that, regardless other components, normalization layers preserve or enhance isometry/ Namely in MLP at initialization (Figure 2 left), and after training (Figure 2, right), and in pretrained transformers (Fig 3).
>
> > - In Section 3.1. what is ment by the sentence "which implies orthogonalization of these results."? Orthogonalization of what?
>
> Assume that $\mathcal{I}(h_\ell) = 1$, then outputs becomes orthogonal to each other since the maximum determinant is obtained for orthogonal vectors. We will add more details about the notion of orthogonality and isometry.
>
>
> > - It would be good to expand the properties of Def 3 a bit more. Now we only have that it is "non-linearity strength for activations". Paper has half a page space for extra content. I would appreciate a few paragraphs of extra content in this context.
>
>
> We acknowledge the need for further elaboration on non-linearity strength. We have a novel experiment confirming that the linearity strengths predicts the convergence of SGD in early epochs, discussed in the general response. We will include additional paragraphs to explain this concept more thoroughly in the revised manuscript, emphasizing its computation, and its influence on training and isometry.
>
>
> > - About Eq. (7), you mention that G_*^l is a sequence that approximates the G^l. From these paragraphs it is hard to see where the sequence is coming from.
>
> Appendix A.1 elaborates on the link between $G_{\*}^l$ and $G^{l}$. The sequence $G_{\*}^l$ refers to the mean-field approximation of the Gram matrix dynamics in the infinitely wide network regime, derived based on pervious mean field approaches in [1,2,3]. Here's the Breakdown of the ideas of why $G_{\*}^l$ is an approximate of $G^l$.  $G^l$ is the actual Gram matrix of the representations at layer $l$. Given the stochastic weights, the dynamics of this sequence is *stochastic*.  The mean-field approximation $G_{\*}^l$ is constructed to represent the average, or mean dynamics, such that conditionally it holds
> $
> E[G^{l+1} | G^l=G^l_\*]=G_\*^{l+1}.
> $
> In contrast to the exact Gram sequence $G^l$ the mean-field Gram sequence $G_\star^l$ is *deterministic.* As we increase the width of the neural network, we observe that dynamics of Gram matrices conditioned on the previous layer becomes highly concentrated around their mean. This implies that at infinite width, evolution of mean-field Gram matrices $G^l_*$ will exactly describe the evolution of Gram matrices $G_l$'s.
>
>
> > ## Questions:
> > - Gap between theory (Theorem 4) and observed in Fig. 4 is quite big. As a future work, is there a clear path for obtaining tighter bound?
>
> This is an excellent question. A likely path to a tighter bound is finding a tighter bound between $\gamma(G)$ (Definition 4, appendix) and isometry gap.
>
> **References**
>
> [1] Samuel S. Schoenholz, Justin Gilmer, Surya Ganguli, and Jascha Sohl-Dickstein. Deep information propagation. In International Conference on Learning Representations, 2017.
>
> [2] Ben Poole, Subhaneil Lahiri, Maithra Raghu, Jascha Sohl-Dickstein, and Surya Ganguli. Exponential expressivity in deep neural networks through transient chaos. Advances in Neural Information Processing Systems, 2016.
>
> [3] Yang, Greg, et al. “A mean field theory of batch normalization.” ICLR (2019).

---

> > ### Comment · Reviewer_B3t5 · 2023-08-18
> >
> > I thank authors for providing a thorough rebuttal and clarification to my concerns. I am quite satisfied with these explanations and I do not have any other issue.

---

> > > ### Author Response · Authors · 2023-08-18
> > >
> > > > I am quite satisfied with these explanations and I do not have any other issue.
> > >
> > > We greatly appreciate the reviewer for their positive evaluation of our work.
> > > We also sincerely thank the reviewer for their time and providing us with their insightful feedback.

---

### Official Review · Reviewer_Qj2n · 2023-07-04

**Soundness:** 3 good
**Presentation:** 2 fair
**Contribution:** 2 fair
**Rating:** 5
**Confidence:** 3

**Summary:**

The paper investigates normalization and non-linear activation functions from the perspective of isometry.

**Strengths:**

- This paper provides interesting analysis into normalization and non-linear activation from the perspective of isometry.
- The paper has provided interesting discussion on future research.

**Weaknesses:**

- Overall the paper looks a bit rushed or not polished. There are many typos for example, for example line 31 "about the the".
- The paper may benefit more from slightly larger scale data. For example, using MNIST could be a good way to improve the paper without too much additional work.

**Questions:**

- One question while reading this paper is, why do we want to preserve the distance and angles between the input data points after mapping them to the feature space? Suppose in image recognition, the desired representation would not preserve the distance and angles in the input data points?
- I think the previous question leads to another question which is also discussed in the paper: why would we want to use isometry to measure and explain layers? Despite many interesting points in the paper, I would suggest the authors add more motivation or probably a few past literature to better justify the motivation of this interesting work.

**Limitations:**

Here are some minor comments:
1. I would recommend the authors polish more the paper. For example, for the figures, I recommend the authors add more captions for readers to better follow and understand the message.
2. I would also suggest improving the presentation a bit. For example, leaving two plots in the last page (page 9) may not be a good idea for the readers.

---

> ### Author Rebuttal · Authors · 2023-08-09
>
> We thank the reviewer for their constructive comments helping us to improve the writing.
>
>
> > The paper may benefit more from slightly larger scale data. For example, using MNIST could be a good way to improve the paper without too much additional work.
>
> To address the reviewer's concern about scale of data and susbstantiate the validtiy of our theorems, we repeated experiments for MNIST (Figure 4), and CIFAR100 (Figure 1) and also Wikidataset for large language models (Figure 2) in the attached PDF to the general response. Let us remark that as theoretical analysis establishes, results holds indepdent from the dataset as long as some mild conditions are met, i.e., samples in the input batch are  linearly independent.
>
>
>
>
> > One question while reading this paper is, why do we want to preserve the distance and angles between the input data points after mapping them to the feature space? Suppose in image recognition, the desired representation would not preserve the distance and angles in the input data points?
>
> Pervious studies demonstrate isometry at initialization leads to a faster convergence of gradient descent (see general response). This is further illuserated in Figure 1 in document attached to the rebuttal respone. In this Figure, activation functions with a stronger isometry bias (i.e. a larger constant $\beta_0$) lead to a faster convergence of stochastic gradient descent. Remarkably, this is only the matter of initialization. The representations will change after training.
>
>
> > I think the previous question leads to another question which is also discussed in the paper: why would we want to use isometry to measure and explain layers? Despite many interesting points in the paper, I would suggest the authors add more motivation or probably a few past literature to better justify the motivation of this interesting work.
>
> We thank the reviewer for raising this lack of clarity on othe motivation and background. several works have attributed the success of normalization and residual/skip connections in training deep neural networks to a faitful signal propagation (also termed dynamic isometry) through deep between layers [1-7]. Here are a few past literature that are directly related to the notion of isometry.
>
> - Dynamic isometry or signal propagation property postulates that in order to ensure a fast training[1,2], the network output must be sensitive to changes in the input. This hypothesis is employed by [4] to train a 10,000 layer CNN by a proper weight initialization, without skip connection or normalization layers.
> - In a similar attempt, [5] develops multiple methods to obtain isometry in transformer architecture that trains with a comparable speed to the model with skip and normalization layers.
> - [6] proposes shaping activations to impose isometry and empirically shows this allows training of very deep neural networks.
> - [7] designs activation functions that impose isometry, hence significantly enhance the training of networks with batch normalization.
>
> We included an experimental result in the general response demonstrating applications of results, and will provide an extensive literature review, motivating our theoretical study.
>
> > # Limitations:
> > Here are some minor comments:
> > - I would recommend the authors polish more the paper. For example, for the figures, I recommend the authors add more captions for readers to better follow and understand the message.
>
> We thank the reviewer for pointing out this lack of clarity in captions. We have added key details regarding the data and experimental procedure to each figure.
>
>
> > - I would also suggest improving the presentation a bit. For example, leaving two plots in the last page (page 9) may not be a good idea for the readers. Overall the paper looks a bit rushed or not polished. There are many typos for example, for example line 31 "about the the".
>
> We will improve the writing. Thank you very much for the typo.
>
> ---
>
> **References**
> [1] Samuel S. Schoenholz, Justin Gilmer, Surya Ganguli, and Jascha Sohl-Dickstein. Deep information propagation. In International Conference on Learning Representations, 2017.
>
> [2] Ben Poole, Subhaneil Lahiri, Maithra Raghu, Jascha Sohl-Dickstein, and Surya Ganguli. Exponential expressivity in deep neural networks through transient chaos. Advances in Neural Information Processing Systems, 2016.
>
> [3] Hadi Daneshmand, Amir Joudaki, and Francis Bach. Batch normalization orthogonalizes representations in deep random networks. Advances in Neural Information Processing Systems 2021.
>
> [4] Xiao, Lechao, et al. "Dynamical isometry and a mean field theory of cnns: How to train 10,000-layer vanilla convolutional neural networks." International Conference on Machine Learning 2018.
>
> [5] He, Bobby, et al. "Deep transformers without shortcuts: Modifying self-attention for faithful signal propagation." arXiv preprint arXiv:2302.10322 (2023)
>
> [6] Zhang, Guodong, Aleksandar Botev, and James Martens. "Deep learning without shortcuts: Shaping the kernel with tailored rectifiers." (2022).
>
> [7] Klambauer, Günter, et al. "Self-normalizing neural networks." Advances in neural information processing systems 30 (2017).

---

> > ### Comment · Reviewer_Qj2n · 2023-08-16
> >
> > I thank the author for providing this thorough rebuttal. The additional experiments on larger datasets seem very interesting and promising.
> >
> > Some concern left (which I think is out of the scope of the paper) is that does faster convergence (in the form of decreasing loss) means that the network is trained better. Previous literature and experiments have also demonstrated that faster drop in loss could lead to overfitting on the training dataset. This could be alleviated by reporting the testing accuracy of experiments in Figure 1 and Figure 2 of the additional material.
> >
> > The biggest concern is still that the paper, in the current form, does not convery the message too well. There needs to be quite substantial amount of work done on polishing the paper (e.g. adding motivation like the author wrote in the rebuttal and changing formats and wording of the work). I can only raise my score to borderline accept and I hope the author tries harder to polish this interesting work in the later versions.

---

> > > ### Author Response · Authors · 2023-08-17
> > >
> > > > The additional experiments on larger datasets seem very interesting and promising.
> > >
> > > We sincerely thank the reviewer increasing their score based on our rebuttal response and appreciating the experiments
> > >
> > > > Previous literature and experiments have also demonstrated that faster drop in loss could lead to overfitting on the training dataset.
> > >
> > > This is an excellent point raised by the reviewer we will subsequently add the test loss plots to the camera ready version.
> > >
> > > > The biggest concern is still that the paper, in the current form, does not convery the message too well. There needs to be quite substantial amount of work done on polishing the paper (e.g. adding motivation like the author wrote in the rebuttal and changing formats and wording of the work). I can only raise my score to borderline accept and I hope the author tries harder to polish this interesting work in the later versions.
> > >
> > > We would like to point that our main contribution is the theoretical results about isometry under normalization and activation layers. None of the rebuttal responses imply any major changes to these parts (except for small typos and additional details in captions). The rebuttal response which was prepared in response to reviewers, which will be added as an additional page. This is accepted under formatting instructions for NeurIPS, as quoted below:
> > >
> > > >  If your submission is accepted, you will be allowed an additional content page for the camera-ready version.
> > >
> > > Please let us know if there are any further questions or points that need clarification.

---

### Official Review · Reviewer_D7v5 · 2023-07-04

**Soundness:** 3 good
**Presentation:** 2 fair
**Contribution:** 3 good
**Rating:** 6
**Confidence:** 3

**Summary:**

The paper is a theoretical analysis in the mean-filed regime the of the second-to-last gram-matrix of MLP. It proves that the presence of layer normalization in conjunction with non-linear activation functions biases the input-output mapping at initialization towards an isometry.

**Strengths:**

This paper presents a novel theoretical contribution that aims to enhance our understanding of the interaction between layer normalization and non-linear activation functions at initialization. The proof relies on general assumptions and provides a justification for the utilization of pre-layer normalized representations in order to prevent training instabilities associated with the collapse of the rank of the input-output gram matrix in a multi-layer perceptron (MLP) network.

**Weaknesses:**

The experimental tests are not well described, for instance the authors do not specify the datasets used neither in the
main text and nor in the captions (see also questions sections).


**Questions:**

1. What kind of input datasets are used in Fig. 2-3-4 ?

2. Why the unit-sphere in d dimension is denoted as $\sqrt d$-sphere and not $d$-sphere?

3. Minor: typo l 146: comma in place of full stop.

**Limitations:**

Not really. Perhaps the main limitation is that the author limit their analysis on MLP architectures. This makes the results of their paper not immediately applicable to transformers (due to the presence of the attention block) which are of much greater practical interest. Maybe this should be stated as a limitation of the current work.

---

> ### Author Rebuttal · Authors · 2023-08-09
>
> We thank the reviewer for their positive feedback and their constructive comments.
>
> > ## Limitations:
> > - Not really. Perhaps the main limitation is that the author limit their analysis on MLP architectures. This makes the results of their paper not immediately applicable to transformers (due to the presence of the attention block) which are of much greater practical interest. Maybe this should be stated as a limitation of the current work.
>
> This is a great suggestion by the reviewer.
> Let us remark that Theorem 1, and corrolaries 2 & 3 hold for any architecture and regardless of the weight distributions. Figure 3 in the `Rebuttal PDF` shows after each Layer Normalization layer in GPT2 (pretrained), the isometry gap decays. Given that the established property of isometry for normalization was deterministic, this is a consequence of these theoretical results.
>
> > ## Weaknesses:
> > The experimental tests are not well described, for instance the authors do not specify the datasets used neither in the main text and nor in the captions (see also questions sections).
>
> We thank the reviewer for pointing out these potential points of ambiguity regarding the experimental section. We will address this by adding detailed explanatioins regarding the experiments.
>
> > ## Questions:
> > - What kind of input datasets are used in Fig. 2-3-4 ?
>
> The input used in these figures wes artificially created to have very low isometry (close to degenerate), so that it highlights the effects of the normalization and activation layers. More specifically, the rows of $d\times n$ are drawn from $N(0, C)$ where $C$ has a highly skewed eigenvalue distribution. Same holds for figures 5,7, 8.  We thank the reviewer for pointing out these missing information in the main text. These details have been added to captions and main text.
>
> > - Why the unit-sphere in d dimension is denoted as $\sqrt{d}$ sphere and not $d$ sphere?
>
> $\sqrt{d}$ denotes the radius of the sphere, but not the dimensionality. The classical definition of LN projects each sample onto the sphere with radius $\sqrt{d}$ as can be seen below:
> \begin{align}
> LN(x) = \frac{(x-\bar x)}{\sqrt{\frac1d \sum_i^d(x_i-\bar x)^2}} \implies \\|LN(x)\\| = \frac{\sqrt{\sum_i^d(x_i-\bar x)^2}}{\sqrt{\frac1d\sum_i(x_i-\bar x)^2}} = \sqrt{d},
> \end{align}
>
> > - Minor: typo l 146: comma in place of full stop.
>
> fixed.

---

> > ### Comment · Reviewer_D7v5 · 2023-08-18
> >
> > I thank the authors for their response.
> > They have addressed my concern regarding the experimental test on transformer architecture. I recommend that they include this explanation in the appendix if the paper will be accepted.
> >
> > I still have a small question about the notation used:
> > I know that $\sqrt d$ is the radius of the sphere where data is projected after LN.
> > My question was about how the authors denote this sphere. From what I know, and as shown in [1] and [2], the $d$-dimensional sphere is usually called the $d$-sphere.
> >
> > [1] https://en.wikipedia.org/wiki/N-sphere
> >
> > [2] https://mathworld.wolfram.com/Sphere.html

---

> > > ### Author Response · Authors · 2023-08-18
> > >
> > >
> > > > the  $d$-dimensional sphere is usually called the $d$-sphere.
> > >
> > > We thank the reviewer for bringing this nuanced notation issue to our attention. Thus we will replace these mentions by "$d$-sphere with radius $\sqrt{d}$"
> > >
> > > > They have addressed my concern regarding the experimental test on transformer architecture.
> > >
> > > We will make sure to include it in the appendix in the revised manuscript.  We again thank the reviewer for this valuable and interesting suggestion. In light of these additional experiments, should you find it fitting, we would be grateful for any potential reconsideration of our score.

---

> > > > ### Comment · Reviewer_D7v5 · 2023-08-18
> > > >
> > > > Thank you for the answer, I appreciate your prompt response.
> > > > I will maintain my current score, which I think already fits the current value of the paper.

---

> > > > > ### Author Response · Authors · 2023-08-18
> > > > >
> > > > > We sincerely thank the reviewer for their valuable and constructive feedbacks and appreciate their positive evaluation of our work.

---

### Official Review · Reviewer_nfuS · 2023-07-04

**Soundness:** 2 fair
**Presentation:** 2 fair
**Contribution:** 2 fair
**Rating:** 5
**Confidence:** 4

**Summary:**

This paper studies the isometry of Gram matrix under the effect of BN, LN and activation at initialization. For BN and LN, some results are obtained. For activation, most are empirical results.

**Strengths:**

Study multiple factors that affect the isometry of Gram matrix.

**Weaknesses:**

1) The novelty of the theory for BN and LN is small.
2) For activation, the conclusion is wekk. It is not strange to see different activation function have different effects. But so what and why?
3) The paper does not justify why study isometry. Its connection to neural networks, such as training speed or generalization, should be demonstrated either empirically or theoretically.

**Questions:**

See weekness.

**Limitations:**

See weekness.

---

> ### Author Rebuttal · Authors · 2023-08-09
>
> We thank the reviewer for their time reviewing our work, providing constructive feedback, and pointing out potential points of confusion.
>
> > This paper studies the isometry of Gram matrix under the effect of BN, LN and activation at initialization. For BN and LN, some results are obtained. For activation, most are empirical results.
>
> Theorem 4 is a theoretical result on activations and one of our main contribution. This theorem is characterizing the isometric properties of activations. The empirical evidence presented are only meant to substantiate and complement this theoretical contribution.
>
> > It is not strange to see different activation function have different effects. But so what and why? The paper does not justify why study isometry. Its connection to neural networks, such as training speed or generalization, should be demonstrated either empirically or theoretically.
>
>
> We directly address these points in supplementary experiments PDF attached to the general response (See figures 1-4).
>
>
> >  The novelty of the theory for BN and LN is small.
>
> The isometry bias of normalization has been the subject of various theoretical studies, including [1,2,3,4, and 5]. While [1-4] establish a local isometry bias in a local neighborhood of certain inputs, we prove the global isometry bias for a wide range of inputs. Furthermore, [5] only proves the global stability for MLP with linear activations and BN. Our theoretical analysis derives a clear connection between non-linear activations and isometry. While pervious results only characterizes this bias for networks with random weights, Theorem 1 proves the isometry result for all networks with normalization layers.
>
>
> ---
>
> **References**
>
> [1] Yang, Greg, et al. "A mean field theory of batch normalization." ICLR (2019).
>
> [2] Li, Mufan, Mihai Nica, and Dan Roy. "The neural covariance SDE: Shaped infinite depth-and-width networks at initialization." Advances in Neural Information Processing Systems (2022).
>
> [3] Samuel S. Schoenholz, Justin Gilmer, Surya Ganguli, and Jascha Sohl-Dickstein. Deep information propagation. In International Conference on Learning Representations, 2017.
>
> [4] Ben Poole, Subhaneil Lahiri, Maithra Raghu, Jascha Sohl-Dickstein, and Surya Ganguli. Exponential expressivity in deep neural networks through transient chaos. Advances in Neural Information Processing Systems, 2016.
>
> [5] Hadi Daneshmand, Amir Joudaki, and Francis Bach. Batch normalization orthogonalizes representations in deep random networks. Advances in Neural Information Processing Systems 2021.

---

> > ### Comment · Reviewer_nfuS · 2023-08-18
> >
> > Thank authors for clarification.

---

> > > ### Author Response · Authors · 2023-08-18
> > >
> > > We thank the reviewer for  increasing their score.
> > >
> > > We would highly appreciate if the reviewer specified which parts of our rebuttal response or their concerns remain unresolved?

---

### Official Review · Reviewer_tNrQ · 2023-07-05

**Soundness:** 2 fair
**Presentation:** 2 fair
**Contribution:** 2 fair
**Rating:** 6
**Confidence:** 3

**Summary:**

This paper studies the isometric properties of a randomly initialized neural network. The authors show layer normalization and proper activation functions can mitigate rank collapse. In addition, they also quantify the normalization bias for different type of normalization layers. They use the Hermite expansion of the activation function to highlight the importance of higher order Hermite coefficients in the bias towards isometry. The paper provides theoretical results and empirical evidence to support its findings and discusses the potential implications for future research on neural network architectures and training algorithms.

**Strengths:**

(+) This paper provides analysis of the isometry properties using the penultimate Gram matrix in neural networks. It complements previous study by analyzing the role of layer normalization (while previous work mainly focus on batch norm). It also proposes a few useful measurement of isometry bias.

(+) The authors provide theoretical results and empirical evidence (mainly figure 2 and figure 3) to show that activation and normalization techniques can bias the Gram matrix towards isometry at initialization, which can improve the training dynamics of deep neural networks.

(+) The paper discusses the potential implications of the findings for future research on neural network architectures and training algorithms, which can inspire new directions for improving the performance and efficiency of deep learning systems.

**Weaknesses:**

(-) I feel this paper is not very coherent between different sections, though possibly because I’m not an expert in this field. I can understand each section but does not find the connection between different sections.

(-) The empirical evidence presented in the paper is quite limited. Only very few activation functions are considered and it also assume the MLPs are of fixed width. The conclusion might be hard to generalize or be helpful to the practice.

(-) The analysis of section 3 and section 4 seems to be two different systems. It does not explain how they are connected and lack a unified theory of them.

(-) There are also a few confusing statements such as the authors show higher He coefficients have negative impact on isometry properties. However, in the abstract, they also state “ highlighting the importance of higher order (>2) Hermite coefficients in the bias towards isometry“, which implies higher order Hermite coefficients help isometry.

Overall, I think this paper has limited applicability and does not inspire new ways on network design (either normalization or activation functions).

**Questions:**

I have some questions regarding the coherence of this paper, for example:
1) what is the motivation of introducing normalization bias in Section 3.2, how does it related to the isometry bias?
2) Is definition 1 the definition for ‘isometry’, or a measure of ‘isometry property’?
3) Is the isometry gap (-logI(M)) ranges from 0 to -\infinity? (It is \infinity on line 100).

I’m also curious how can the insights from this paper be used to design more efficient and effective deep learning systems? What is mainly the assumptions you make that present challenges of applying these insights in practice?

**Limitations:**

This paper is unlikely to have potential negative societal impact.

---

> ### Author Rebuttal · Authors · 2023-08-09
>
> We thank the reviewer for their time reviewing our work, providing constructive feedback, and pointing out potential points of confusion.
>
> *Detailed responses:*
> >  There are also a few confusing statements such as the authors show higher He coefficients have negative impact on isometry properties. However, in the abstract, they also state “ highlighting the importance of higher order (>2) Hermite coefficients in the bias towards isometry“, which implies higher order Hermite coefficients help isometry.
>
> Theoretical result aligns with the statement in the abstract. There is a subtle difference between isometry ($\mathcal{I}$) and isometry gap ($-\log\mathcal{I}$): The negative logorithmic of isometry is the isometry gap that quantifies how far a matrix is from isometry. Thus Figure 5 shows that  $He_2, He_3,$ lead to a decay of isometry gap, which implies higher isometric properties. This is consistent with our main result and the summary stated in abstract. Extremium values of both isometry and isometry gap are elaborated in lines 97-101 of the main text: isometry lies between $0$ (degenerate matrix) and $1$ (perfect isometry, ie, orthogonal matrix), while isometry gap lies between $-\infty$ (degnerate) and $0$ (orthogonal matrix, ie, perfect isometry). According to Theorem 4, the higher order Hermite cofficient imposes a faster decay in isometry gap, hence a higher faster convergence for isometry to 1. For example, there is no decay in $-\log \mathcal{I}$ for linear activations. Therefore, linear activation does not impose isometry which is experimentally substantiated in Figure 3.
>
> > The empirical evidence presented in the paper is quite limited. Only very few activation functions are considered
>
> To address the reviewer's concerns about limitations of our theory, we have expanded empirical results to include activations `Identity, ReLU, PReLU, Tanh, SiLU(Swish), ELU, GELU, SELU.` Figure 2 in the PDf attached to the general response depicts the isometry across layers of an MLP for various activations, along with their $\beta_0$ value. As can be seen, the value of $\beta_0$ accurately predicts the isometry gap decay.
>
>
> > I feel this paper is not very coherent between different sections, though possibly because I’m not an expert in this field. I can understand each section but does not find the connection between different sections. The analysis of section 3 and section 4 seems to be two different systems. It does not explain how they are connected and lack a unified theory of them.
>
> Both results are characterizing isometry across the layers of MLP. While Section 3 focus on a single normalization, Section 4 investigates the isometry for MLPs with layer normalization and non-linear activations. We will add an outline before section 3 to highlight the connection between different sections. In Figure 2 attached to the general response, we illustrate how these two results connects and when we can invoke results in these sections for deep neural networks. We will elaborate on the coherence .
>
> > and it also assume the MLPs are of fixed width. The conclusion might be hard to generalize or be helpful to the practice.
>
> The assumption that MLP has fixed width is merely to ease the notation. In fact, one can readily extend the results to an MLP with variable widths across layersas long as these widths are sufficiently large, ie., $1/\sqrt{width}$ is small). To substantiate this, we have added experiments with variable width (see PDF attached to general response, Figure 4).
>
>  > Overall, I think this paper has limited applicability and does not inspire new ways on network design (either normalization or activation functions).  I’m also curious how can the insights from this paper be used to design more efficient and effective deep learning systems? What is mainly the assumptions you make that present challenges of applying these insights in practice?
>
> The general response demonstrates the following applications:
> - **Predicting training speed with $\beta_0$**: The non-linearity coefficient $\beta_0$ is a strong indicator for training convergence speed (see general response and Figure 1 of attached PDF to the general response)
> - **Explaining isometry bias of activations** Theorem 4 and non-linearity strength $\beta_0$ accurately predict the isometry of activations for a wide range of activations  (see Figure 2 of attached PDF to the general response)
> - **Explaining isometry bias of layer normalization in MLPs and transformers** Theorem 1 and corollary 2 prove the isometry-enhancing properties of  LayerNorm in MLPs (Figure 2, attached PDF) and transformers (Figure 3, attached PDF)
>
> > what is the motivation of introducing normalization bias in Section 3.2, how does it related to the isometry bias?
>
> They both refer to the "isometry bias of normalization." To avoid creating the impression that they are separate concepts, we will replace occurrences with "isometry bias of normalization" in the revised manuscript.  We introduced the normalization bias to show normalization layers impose isometry across the layers of neural networks. When normalization bias is significantly greater than zero, normalization layers constantly increases the isometry across the layers. Figure 2 and 3 illustrate the consequence of having high normalization bias. As we observe in Figure 3, having high normalization bias (in Figure 2) concludes $-\log I$ decays across the layers of neural networks.
>
> > Is definition 1 the definition for ‘isometry’, or a measure of ‘isometry property’?
>
> It is the definition for isometry. Notably, the terms of isometry is slightly different from isometric maps that preserves the distances.
>
> > Is the isometry gap $(-\log I(M))$ ranges from 0 to $-\infty$? (It is \infinity on line 100).
>
> *Isometry gap ranges between $0$ and $\infty$:* Since isometry is smaller than $1,$ its logarithm is non-positive $\log I(M)\le 0$. hence Isometry gap  is non-negative $-\log I(M)\ge 0$.

---

> > ### Comment · Reviewer_tNrQ · 2023-08-16
> >
> > Thank you for your detailed reply.
> >
> > My question about 1) inconsistency of the effect He coefficients in experiments and abstract; 2) MLP of fixed width; 3) limited activation functions are addressed.
> >
> > I have two follow-up questions:
> >
> > Can you clarify the relationship between isometry gap, normalization bias, isometry bias, $-\log(I)$ ? In my understanding: isometry bias including isometry bias of normalization and isometry bias of activation. Is this correct? I have this question because figure 2 and 3 both show Isometry bias of normalization while the y-axis is "normalization bias" and "$-log(I)$" separately. What is the difference between the two figures?
> >
> > Another question is about identity activation function. I thought identity function would preserve the isometry property (it preserve the distance), but this seems to be contradict with Figure 3. Is this because identity actiavtion is not helpful on recovering the isometry gap (although it does not make it worse) while other activations are?

---

> > > ### Author Response · Authors · 2023-08-17
> > >
> > > We highly appreciate the reviewer's response to our rebutal response.
> > >
> > > > Another question is about identity activation function. I thought identity function would preserve the isometry property (it preserve the distance), but this seems to be contradict with Figure 3. Is this because identity actiavtion is not helpful on recovering the isometry gap (although it does not make it worse) while other activations are?
> > >
> > > Yes the reviewer is correct. We thank the reviewer for making this nuanced observation. Please note that there is a distinction between preserving isometry (which is what identity achieves), and improving isometry (which is what non-linear activations achieve). We can see this in Theorem 4, equation (5), that isometry gap decays exponentially with depth $\ell$ with rate $\exp(-\ell \log\beta_0)$. Thus, for any activation with non-linear components $\beta_0 > 1,$ we have $\log\beta_0 > 0,$ while identity activation we have $\log\beta_0 = 0$, which affirms your observation.
> > >
> > >
> > > > Can you clarify the relationship between isometry gap, normalization bias, isometry bias, ? In my understanding: isometry bias including isometry bias of normalization and isometry bias of activation. Is this correct?
> > >
> > >
> > > The reviewer is correct in assuming that various components of neural networks such as normalization and activation layers, influence isometry. Let us clarify the definition of these terms and where they are defined in the main text:
> > >
> > >
> > > | Term | Definition| Where defined | Range |
> > > | -------- | -------- | -------- | ----- |
> > > | Isometry  | $\mathcal{I} = \frac{G.M.(eigs)}{A.M.(eigs)}$    | Table 1 | $[0,1]$ |
> > > | Isometry gap  | $-\log\mathcal{I}$     | Table 1 | $[0,\infty]$|
> > > | Normalization bias | $\frac{Var(\text{norms)}}{\text{mean}(\text{norms})^2}$ | equation 4 & 5 | $[0,\infty]$ |
> > >
> > > - Isometry bias of normalization: According to Theorem 1, a larger normalization bias causes a larger increase in isometry (or decrease in isometry gap) after passing through the normalization layer:o
> > > $$
> > > \mathcal{I}(\text{post-normalization Gram}) \ge \mathcal{I}(\text{pre-normalization Gram})(1+\text{normalization bias})
> > > $$
> > >
> > > - Isometry bias of activation: According to theorem 4, isometry gap of MLP with activation non-linearity strength $\beta_0$ decays exponentially with rate $1/\beta_0$:
> > > $$
> > > -\log\mathcal{I}(\text{Gram matrix of layer $\ell$}) \lesssim \exp(-\ell \log\beta_0) = (1/\beta_0)^\ell
> > > $$
> > >
> > > > I have this question because figure 2 and 3 both show Isometry bias of normalization while the y-axis is "normalization bias" and "$-\log \mathcal{I}$" separately. What is the difference between the two figures?
> > >
> > > Figure 2 is plotting normalization bias, as defined in equation (4) and (5), but, in Figure 3 the y-axis is showing the isomtery gap, as defined in Table 1.
> > >
> > > > Overall, I think this paper has limited applicability and does not inspire new ways on network design ...
> > >
> > > We wonder if the reviewer finds our responses regarding applicability convincing or not?

---

> > > > ### Comment · Reviewer_tNrQ · 2023-08-17
> > > >
> > > > Thank you for your reply. I think all the previous discussions (as well as rebuttal to other reviewers) make the paper much more clear. These should be incorporated in the revision.
> > > >
> > > > I think the applicability increases compared to the initial version. I'm still skeptical (but not negative) on its impact to practical network design but this is not a decision factor since I understand the focus is on theory and practical verfications is beyond the scope.
> > > >
> > > > I'm willing to increase my score to weak accept.

---

> > > > > ### Author Response · Authors · 2023-08-18
> > > > >
> > > > > > I'm willing to increase my score to weak accept.
> > > > >
> > > > > We sincerely thank the reviewer for increasing their score, as well as for taking the time to give us constructive feedback.
> > > > >
> > > > > > I think all the previous discussions (as well as rebuttal to other reviewers) make the paper much more clear. These should be incorporated in the revision. I think the applicability increases compared to the initial version.
> > > > >
> > > > > We wholeheartedly agree with the reviewer that the review process dramatically improved the practical implications of our theoretical results. We will make sure to incorporate them in the final version.

---

### Author Rebuttal · Authors · 2023-08-09

We appreciate constructive reviews that helped us to improve the paper. Let us recount our main contributions by showing excerpts from the reviews.

This excerpt from `B3t5`'s captures the main part of contribution:
> A study of layernorm and Gramm matrix isometry is presented in both theory and emprical results... Application of layernorm after activation seems to be a correct strategy when we are interested to isometry of output Gramm matrix.

This view is shared by reviewer `D7v5` highlighting applications of our result to training instabilities in deep neural networks
> The proof relies on general assumptions and provides a justification for the utilization of pre-layer normalized representations in order to prevent training instabilities associated with the collapse of the rank of the input-output gram matrix in a multi-layer perceptron (MLP) network.

Furthermore, reviewer `Qj2n` views our analysis interesting:
> This paper provides interesting analysis into normalization and non-linear activation from the perspective of isometry.


Reviewers have also raised some concerns which we have  addressed with experiments and references:


## Background and motivation for isometry in the literature.
Several studies have attributed the success of normalization and residual/skip connections in training deep neural networks to a faitful signal propagation (also termed dynamic isometry) through deep between layers [1-7]. Related literature demonstrate the important role of isometry in training stabilizing the training of deep neural networks:

- Dynamic isometry or signal propagation property postulates that in order to ensure a fast training[1,2], the network output must be sensitive to changes in the input. This hypothesis is employed by [4] to train a 10,000 layer CNN by a proper weight initialization, without skip connection or normalization layers.

- [5] develops multiple methods to obtain isometry in transformer architecture that trains with a comparable speed to the model with skip and normalization layers.

- [6] proposes shaping activations to impose isometry and empirically shows this allows training of very deep neural networks.
- [7] designs activation functions that impose isometry, hence significantly enhance the training of networks with batch normalization.

## A  supplementary experiment
To elaborate on applications of our theoretical results, we supplement our findings with an experiment showing that:

 **Non-linearity strength $\beta_0$ is a strong predictor for training performance**

In Figure 1 of the attached PDF, we compare the convergence rates of stochastic gradient descent on a Multi-Layer Perceptron (MLP) with various activations for which $\beta_0$ (as defined in Eq. 6) is monotonically increasing. We observe activation functions with higher values of $\beta_0$ (as per Eq. 6) result in accelerated convergence for stochastic gradient descent. It is important to note that $\beta_0$ governs the isometry bias of the activation, as proven in Theorem 4. Consequently, our findings carry direct practical implications, shedding light on the popularity of certain activation functions such as ReLU in deep learning.

## Generality of our results for MLPs and transformers
Several of the questions raised are related to generality and broader applicability of our theoretical results. In response, we have provided the following explanatioins and clarifications:
- **Isometry of LayerNorm after training**: Theorem 1 and corollary 2 are valid in any setting, regardless of the architecture and the distribution of weights This is demonstrated in Figure 2 of the attached PDF
- **Isometry of LayerNorm in transformers** Isometry-enhancing property of LayerNorm, as predicted by corollary 2, is architecture-independent. Figure 3 of the attached PDF shows isometry bias of normalization layers for a pretrained GPT2 architecture.
- **Broad range of activations in MLPs** As demonstrated in Figure 2 of the attached PDF, the non-linearity strength


We hope that with these additional clarifications and new empirical evidence, the reviewer's concerns are addressed.


---

**References**

[1] Samuel S. Schoenholz, Justin Gilmer, Surya Ganguli, and Jascha Sohl-Dickstein. Deep information propagation. In International Conference on Learning Representations, 2017.

[2] Ben Poole, Subhaneil Lahiri, Maithra Raghu, Jascha Sohl-Dickstein, and Surya Ganguli. Exponential expressivity in deep neural networks through transient chaos. Advances in Neural Information Processing Systems, 2016.

[3] Hadi Daneshmand, Amir Joudaki, and Francis Bach. Batch normalization orthogonalizes representations in deep random networks. Advances in Neural Information Processing Systems 2021.

[4] Xiao, Lechao, et al. "Dynamical isometry and a mean field theory of cnns: How to train 10,000-layer vanilla convolutional neural networks." International Conference on Machine Learning 2018.

[5] He, Bobby, et al. "Deep transformers without shortcuts: Modifying self-attention for faithful signal propagation." arXiv preprint arXiv:2302.10322 (2023)

[6] Zhang, Guodong, Aleksandar Botev, and James Martens. "Deep learning without shortcuts: Shaping the kernel with tailored rectifiers." (2022).

[7] Klambauer, Günter, et al. "Self-normalizing neural networks." Advances in neural information processing systems 30 (2017).

---

> ### Comment · Area_Chair_PQVn · 2023-08-18
> **Thank you for the rebuttal**
>
> Dear authors,
>
> thank you for providing a rebuttal. Some of the reviewers have already replied, so this is just to let you know that I am in contact with the remaining ones as well.
>
> Best,
> Your AC

---

### Decision · Program_Chairs · 2023-09-21

**Decision:**

Accept (poster)

**Comment:**

This paper considers the Gram matrix at the penultimate layer, and it proves that layer normalization biases such matrix towards an isometry. The rate of convergence is quantified via the Hermite expansion of the activation function, and it is exponential in the network depth. As it is typical of this line of work, the analysis is carried out at initialization. Empirical results are presented to validate the theory.

The authors have provided a comprehensive rebuttal addressing the concerns of the reviewers, who have now reached a consensus towards accepting this paper (although with different levels of support). After my own reading of the reviews, rebuttal and paper, I agree with this view and find the main results valuable and insightful. Hence, I am happy to recommend acceptance. I warmly encourage the authors to incorporate their rebuttal in the final version.